# PerceptionCLIP: Visual Classification by Inferring and Conditioning on Contexts

**Bang An**[1*]     **Sicheng Zhu**[1*]     **Michael-Andrei Panaitescu-Liess**[1]
**Chaithanya Kumar Mummadi**[2]     **Furong Huang**[1]

[1]University of Maryland, College Park     [2]Bosch Center for Artificial Intelligence

## Abstract

Vision-language models like CLIP are widely used in zero-shot image classification due to their ability to understand various visual concepts and natural language descriptions. However, how to fully leverage CLIP's unprecedented human-like understanding capabilities to achieve better performance is still an open question. This paper draws inspiration from the human visual perception process: when classifying an object, humans first infer contextual attributes (e.g., background and orientation) which help separate the foreground object from the background, and then classify the object based on this information. Inspired by it, we observe that providing CLIP with contextual attributes improves zero-shot image classification and mitigates reliance on spurious features. We also observe that CLIP itself can reasonably infer the attributes from an image. With these observations, we propose a training-free, two-step zero-shot classification method `PerceptionCLIP`. Given an image, it first infers contextual attributes (e.g., background) and then performs object classification conditioning on them. Our experiments show that `PerceptionCLIP` achieves better generalization, group robustness, and interpretability. Our code is available at `https://github.com/umd-huang-lab/perceptionCLIP`.

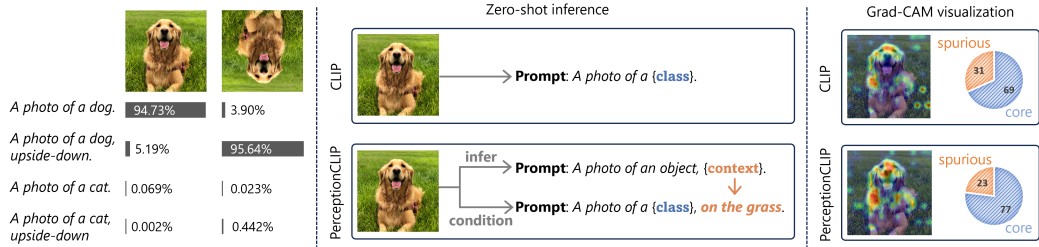

Figure 1: **(Left)**: CLIP co-relates natural language descriptions of contextual attributes with visual cues (*orientation: upside-down*). **(Center)**: Unlike CLIP's standard zero-shot inference that uses fixed template(s) for class name retrieval, our method first infers contextual attributes (*background: on the grass*) using CLIP and then let CLIP predicts the class conditioned on the inferred contextual attributes. Here, background and orientation are both examples of contextual attributes. **(Right)**: Grad-CAM visualization illustrates that our method focuses more on core features (*on the dog*) and is less distracted by spurious features (*grass background*) when performing the object classification.

# 1 Introduction

CLIP (Contrastive Language-Image Pretraining by Radford et al. (2021)) is a foundational Vision-Language Model (VLM) that connects the fields of vision and natural language. By pretraining on 400 million image-caption pairs, CLIP can associate various visual concepts with their corresponding natural language descriptions, making it the foundation for numerous other vision-language models

---

*Equal contribution

(Zhu et al., 2023; Liu et al., 2023; Dai et al., 2023; Li et al., 2023b), diffusion models (Ramesh et al., 2022; Rombach et al., 2022), and semantic segmentation models (Kirillov et al., 2023). This remarkable understanding capability of CLIP is significant for zero-shot classification (Larochelle et al., 2008), enabling open-ended image classification via natural language without training. This capability also addresses many challenging tasks with limited or no downstream data, such as model deployment in the wild (Li et al., 2023a), medical image classification (Wang et al., 2022) and satellite object recognition (Ramaswamy et al., 2023).

Although CLIP shows strong potential in zero-shot classification, current methods treat image classification as a text retrieval task and lack systematic investigation into the text prompts used. This leads to sub-optimal generalization (Radford et al., 2021), reliance on spurious features (Yang et al., 2023), biased predictions (Agarwal et al., 2021; Chuang et al., 2023), and lack of interpretability (Zhou et al., 2022b; Menon & Vondrick, 2022). For example, Radford et al. (2021) uses a basic template "*a photo of a {class name}*" to identify the most relevant class for an image, much less informative than the image captions used during pretraining (see examples in Table 9). Another method, prompt ensembling (Radford et al., 2021), employs 80 crafted templates for better generalization. Nevertheless, it remains unclear whether these templates are optimal and why they are effective. By treating zero-shot classification simply as a class name retrieval problem, these methods potentially waste the capability of CLIP to understand both class-specific features and class-independent attributes such as background and orientation (referred to as contextual attributes in this paper).

Given CLIP's unprecedented human-like vision and language understanding, a natural idea is to draw inspiration from human visual perception. Classic neuroscience (Kandel et al., 2013) describes human visual perception as a three-tiered, context-dependent process: first discerning basic visual attributes like color and orientation, then analyzing scene layout and distinguishing foreground from background, and finally recognizing objects (see details in Appendix C). For example, when humans classify objects in images, we unconsciously acquire contextual attributes like the background and orientation, and in the case of an upside-down image (Figure 1 left), we first infer that the image is rotated and then calibrate our classification accordingly. This hierarchical and context-dependent process contrasts with existing classification methods, which overlook contextual attributes.

Building on this insight, we propose a zero-shot classification method called `PerceptionCLIP`, which emulates a crucial part of human visual perception — inferring and conditioning on the contextual attributes — resulting in improved generalization, reduced reliance on spurious features, better group robustness, and interpretability. Our contributions are as follows:

▷ **(1)** We prepare CLIP for perception by structuring CLIP-understandable text prompts with contextual attributes and introducing an attribute-aware CLIP score to approximate essential conditional probabilities for perception emulation.

▷ **(2)** Through two proof-of-concept investigations, we reveal that conditioning on ground-truth contextual attributes improves CLIP's zero-shot classification and mitigates reliance on spurious features. Moreover, CLIP has the ability to infer contextual attributes by itself.

▷ **(3)** Based on the observations, we propose `PerceptionCLIP`. Given an image, as shown in Figure 1, it first employs CLIP to infer contextual attributes. Then, it uses CLIP to infer the class conditioned on the attributes by incorporating the descriptions of the inferred attributes into the prompt. This two-step inference resembles the concept of chain-of-thoughts in language models.

▷ **(4)** We empirically demonstrate that `PerceptionCLIP` excels in both standard generalization and group robustness, exhibiting improved interpretability. For generalization, it consistently outperforms baselines that use simple templates and prompt ensembles on 11 datasets. For example, it provides a near $5\%$ accuracy gain on the EuroSAT dataset. For group robustness, it reduces the gap between average accuracy and worst group accuracy by $19\%$ on the Waterbirds dataset and $7\%$ on the CelebA dataset with ViT-L/14, showing less reliance on spurious features.

## 2 RELATED WORK

Due to CLIP's ability to understand finer-grained visual concepts beyond classes, some work also leverages external knowledge to augment prompts. For example, Menon & Vondrick (2022); Pratt et al. (2022); Mao et al. (2022); Feng et al. (2023) use large language models to generate class-specific descriptions, resulting in prompts like "a photo of a hen, which has two legs". Novack et al. (2023)

use class hierarchies to generate sub-classes for each parent class and aggregate model predictions on all sub-classes to get a final prediction. Udandarao et al. (2023) use class names to retrieve and maintain some auxiliary data to help downstream classification. In contrast, our method addresses class-independent attributes (i.e., contextual attributes) such as background and orientation, whose comprehension by CLIP is not well-known. These attributes are also combinatorial, potentially covering more aspects of an image than class-specific attributes. Moreover, we can still leverage contextual attributes (e.g., gender, age) when class-specific attributes are hard to articulate, as in the hair-color classification tasks on CelebA. We defer more related work to Appendix A.

## 3 Preliminaries

**Notation.** We use uppercase letters to denote random variables and lowercase letters to denote their realizations. For a random variable $Z$, we use $p_Z(z)$ to denote its probability mass or density function, and omit the subscript $Z$ when the function's meaning can be inferred from the input notation $z$.

**Pretraining of CLIP.** CLIP is pretrained on web-scale image-caption pairs, using a contrastive loss to learn good image and text representations in a shared space, aiming to correctly associate images and their textual descriptions. The captions in the pretraining data (as shown in Table 9) typically describe not only the object's class but also contextual attributes like color, style, and background.

**Zero-shot classification.** After pretraining, Radford et al. (2021) use a universal prompt template, represented by an **annotation function** $\alpha(y) = $ *"a photo of a {class name of y}"*, that takes the **class index** $y$ as the input and outputs a text that only describes the class. For any **image** $x$ in the image space $\mathcal{X}$ and $y$ in the class set $\mathcal{Y}$, the CLIP model serves as a score function $\texttt{CLIP}_1 : \mathcal{Y} \times \mathcal{X} \to \mathbb{R}$ via

$$\texttt{CLIP}_1(y; x) \triangleq \langle \phi_I(x), \quad \phi_T(\alpha(y)) \rangle, \tag{1}$$

computing a similarity score (within $[-1, 1]$) between the image and text through inner products of their representations produced by **image encoder** $\phi_I$ and the **text encoder** $\phi_T$. The subscript '1' in '$\texttt{CLIP}_1$' indicates that only one textual template is used. Then, given an image $x$, the method predicts the class $\hat{y} \in \mathcal{Y}$ as the one with the highest $\texttt{CLIP}_1$ score, $\hat{y} = \arg\max_{y \in \mathcal{Y}} \texttt{CLIP}_1(y; x)$.

In addition, Radford et al. (2021) propose prompt ensembling, which ensembles 80 manually-designed templates $\{\alpha_i\}_{i=1}^{80}$, such as *'a bad photo of a {class name of y}'* and *'a sculpture of a {class name of y}'*, and replace $\texttt{CLIP}_1$ with the following $\texttt{CLIP}_{80}$ score for inference. Prompt ensembling involves some contextual attributes in the templates, but it is ad-hoc and lacks a systematic analysis.

$$\texttt{CLIP}_{80}(y; x) \triangleq \left\langle \phi_I(x), \quad \frac{\frac{1}{80} \sum_{i=1}^{80} \phi_T(\alpha_i(y))}{\left\| \frac{1}{80} \sum_{i=1}^{80} \phi_T(\alpha_i(y)) \right\|} \right\rangle. \tag{2}$$

## 4 Preparing CLIP for Perception

### 4.1 Structuring and Describing Contextual Attributes

**Contextual attributes as generative factors.** We consider contextual attributes as generative factors that contribute to the data generation process. Specifically, let $Y$ denote the underlying object class (e.g., *dog*) that takes values in the class set $\mathcal{Y}$. Let each $Z_i$ ($1 \le i \le m$) denote a certain **contextual attribute** of the object (e.g., *orientation*) that takes values in the contextual attribute set $\mathcal{Z}_i$ (e.g., {*upright*, *upside-down*, *rotated*}) and is causally independent (Pearl, 2009) of the object class $Y$. Then, we consider an image $X$ to be generated as $Y \to X \leftarrow \{Z_i\}_{i=1}^m$.

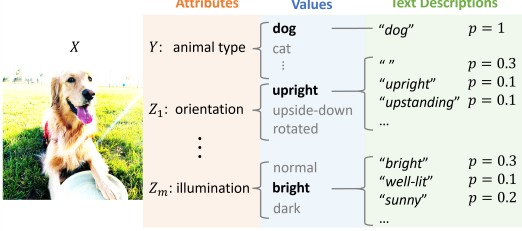

Figure 2: Illustration of contextual attributes, their symbolic discrete values, and the possible textual descriptions mapped by the annotation function.

**Textual descriptions for contextual attributes.** While CLIP requires semantic text, generative factors are often symbolized discrete values, thus creating a gap. It is negligible for the objects' classes since class names are descriptions with no ambiguities. However, the textual descriptions

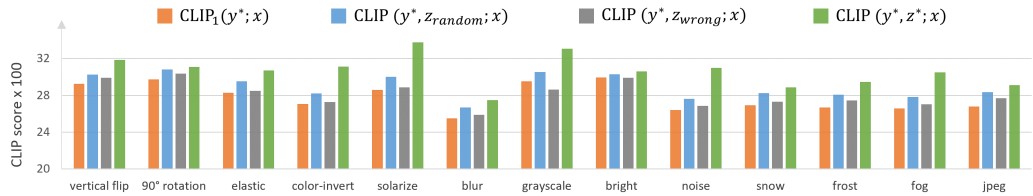

Figure 3: Evaluating CLIP scores on ImageNet with different transformations altering the contextual attributes. The attribute-aware `CLIP` score gives higher scores for correctly matched image-attribute pairs (green) while giving lower scores for mismatched pairs (grey) and random pairs (blue), confirming CLIP's understanding of our contextual attribute descriptions. `CLIP` score measures the similarity between images and contextual attributes, while the original CLIP score (orange) is attribute-agnostic.

of the contextual attributes are vague. Taking upright images as an example, people may use terms like "upright," "upstanding," or no description since it is a common direction. To bridge this gap and translate discrete values into CLIP-readable text, we introduce a specific **annotation function** $\alpha : \mathcal{Z} \to \mathcal{P}(\text{text})$, which maps a symbolic discrete value in $\mathcal{Z}$ to a distribution over natural language textual descriptions. Figure 2 illustrates some examples. An ideal annotation function models people's preferences when captioning images. We form the final image description using the **concatenation operation** $\oplus$. This operation results in a new description distribution $\alpha(y) \oplus \alpha(z_1) \oplus \alpha(z_2) \oplus ...$ where attributes' descriptions are concatenated together and separated by commas. For example, when $y, z_1, z_2$ represent "*dog*," "*upright*," and "*bright*" respectively, the concatenation $\alpha(y) \oplus \alpha(z_1) \oplus \alpha(z_2)$ yields the description "*a photo of a dog, upright, bright,*" or "*a photo of a dog, sunny,*" etc.

## 4.2 Connecting Conditional Probabilities with CLIP Score

**Attribute-aware `CLIP` score.** Existing CLIP score is agnostic of contextual attributes and thus cannot approximate conditional probabilities that are attribute-dependent. Therefore, we define a new score function $\texttt{CLIP} : \mathcal{Y} \times \mathcal{Z}_1 \times \cdots \times \mathcal{Z}_m \times \mathcal{X} \to \mathbb{R}$:

$$\texttt{CLIP}(y, z_1, \ldots, z_m; x) \triangleq \left\langle \phi_I(x), \quad \frac{\mathbb{E} \ \phi_T\big(\alpha(y) \oplus \alpha(z_1) \oplus \cdots \oplus \alpha(z_m)\big)}{\|\mathbb{E} \ \phi_T\big(\alpha(y) \oplus \alpha(z_1) \oplus \cdots \oplus \alpha(z_m)\big)\|} \right\rangle. \quad (3)$$

It takes contextual attributes $z_i$s as additional inputs, describes them internally alongside the class through the annotation function $\alpha(z_i)$, and calculates the similarity with the image in the embedding space. The expectation is taken over the randomness of the descriptions of contextual attributes. The defined `CLIP` score captures the contextual attributes and behaves like an energy function (LeCun et al., 2006): it is high for correctly matched image-attribute pairs while low for mismatched ones. More formally, when $(y^*, z_1^*, \ldots, z_m^*)$ are the ground-truth class and attributes that generate image $x^*$ whereas $(y, z_1, \ldots, z_m)$ are some arbitrary class and attributes,

$$\texttt{CLIP}(y^*, z_1^*, \ldots, z_m^*) \geq \texttt{CLIP}(y, z_1, \ldots, z_m), \quad \forall\, y \in \mathcal{Y}, \quad \forall\, z_i \in \mathcal{Z}_i, \quad \forall\, 1 \leq i \leq m. \quad (4)$$

Figure 3 and 5 empirically verified this property (see Appendix E.1 for details). Given the pretraining process, this observation is not surprising since it encourages high scores for correctly matched image-caption pairs where the caption describes not only the class but also the contextual attributes.

**Approximating conditional probabilities.** With the energy-function-like `CLIP` score, we approximate the conditional probabilities. Specifically (in Table 1 and Appendix D), we approximate **(1) the joint conditional probability** $p(y, z_1, \ldots, z_m | x)$, which measures the likelihood of an object class and some contextual attributes occurring together given the image, requiring only exponentiation and normalization. Based on it, we derive the rest two using the law of total probability. **(2) the conditional probability** $p(y | z_1, \ldots, z_m, x)$, which measures the probability of an object class given both the image and

Table 1: Conditional probabilities. $x, y$, and $z$ denote image, class, and contextual attributes. $z$ denotes $(z_1, \ldots, z_m)$ for simplicity.

| Probability | Approximation |
|---|---|
| $p(y, z\|x)$ | $\dfrac{e^{\texttt{CLIP}(y,z;x)}}{\sum_y \sum_z e^{\texttt{CLIP}(y,z;x)}}$ |
| $p(y\|x, z)$ | $\dfrac{e^{\texttt{CLIP}(y,z;x)}}{\sum_y e^{\texttt{CLIP}(y,z;x)}}$ |
| $p(z\|x)$ | $\dfrac{\sum_y e^{\texttt{CLIP}(y,z;x)}}{\sum_z \sum_y e^{\texttt{CLIP}(y,z;x)}}$ or $\dfrac{e^{\texttt{CLIP}(z;x)}}{\sum_z e^{\texttt{CLIP}(z;x)}}$ |

Table 2: Classification accuracy (%) on ImageNet. We apply the left-side image transformations to alter the corresponding attributes. Different methods condition on different values of the contextual attributes. Conditioning on correct or self-inferred attribute values improves accuracy the most.

| Contextual attribute | Accuracy | | | | |
|---|---|---|---|---|---|
| | w/o $z$ | w/ random $z$ | w/ wrong $z$ | w/ correct $z$ | w/ self-infer $z$ |
| vertical flip | 51.17 | 52.02 (↑0.85) | 52.19 (↑1.02) | 52.48 (↑1.31) | 52.54 (↑**1.37**) |
| 90° rotation | 57.02 | 58.38 (↑1.36) | 58.23 (↑1.21) | 58.75 (↑**1.73**) | 58.30 (↑1.28) |
| elastic-transform | 48.66 | 48.45 (↓0.21) | 48.75 (↑0.09) | 48.89 (↑0.23) | 49.00 (↑**0.34**) |
| color-invert | 35.29 | 36.12 (↑0.83) | 35.89 (↑0.60) | 36.72 (↑1.43) | 36.80 (↑**1.51**) |
| solarize | 49.79 | 49.74 (↓0.05) | 50.20 (↑0.41) | 50.49 (↑0.70) | 50.54 (↑**0.75**) |
| blur | 38.86 | 39.65 (↑0.79) | 39.21 (↑0.35) | 39.92 (↑**1.06**) | 39.80 (↑0.94) |
| grayscale | 59.51 | 59.67 (↑0.16) | 59.48 (↓0.03) | 59.98 (↑0.47) | 60.04 (↑**0.53**) |
| bright | 60.81 | 62.04 (↑**1.23**) | 60.94 (↑0.13) | 61.41 (↑0.60) | 61.28 (↑0.47) |
| noise | 14.16 | 14.88 (↑0.72) | 14.75 (↑0.59) | 15.66 (↑1.50) | 15.68 (↑**1.52**) |
| snow | 33.09 | 32.94 (↓0.15) | 33.56 (↑0.47) | 34.50 (↑**1.41**) | 34.33 (↑1.24) |
| frost | 31.08 | 31.91 (↑0.83) | 31.76 (↑0.68) | 32.63 (↑1.55) | 32.81 (↑**1.73**) |
| fog | 37.61 | 38.40 (↑0.79) | 38.00 (↑0.39) | 39.31 (↑1.70) | 39.34 (↑**1.73**) |
| jpeg | 33.67 | 34.80 (↑1.13) | 35.11 (↑1.45) | 35.39 (↑1.72) | 35.47 (↑**1.80**) |
| average | - | ↑0.64 | ↑0.57 | ↑1.16 | ↑**1.17** |

the contextual attributes, which is our main inference objective. **(3) the conditional probability $p(z_1, \ldots, z_m | x)$**, measures the likelihood of some contextual attributes given the image and is used for inferring the contextual attributes. We provide two approximations, referred to as *ClassAttr* (left) and *PureAttr* (right). The textual description corresponding to CLIP$(y, z; x)$ in *ClassAttr* is "*a photo of a {class name of y}, {description of z}*," while the description corresponding to CLIP$(z; x)$ in *PureAttr* is "*a photo of an object, {description of z}*" with a word like "object" substituting all classes.

# 5    Contextual Attributes are Helpful and Inferable

This section presents proof-of-concept experiments showing that emulating human perception through conditional inference on contextual attributes improves zero-shot classification. Additionally, such improvement does not require ground-truth attributes, as CLIP itself can infer attributes reasonably.

## 5.1    Conditioning on Contextual Attributes is Helpful

We first evaluate if conditioning on the ground-truth contextual attributes improves the zero-shot classification accuracy. Given an image $x$, the most likely class is $\hat{y} = \arg\max_y \ p(y|x, z^*)$ with:

$$\arg\max_y \ p(y|x, z^*) = \arg\max_y \ \frac{e^{\text{CLIP}(y, z^*; x)}}{\sum_y e^{\text{CLIP}(y, z^*; x)}} = \arg\max_y \ \text{CLIP}(y, z^*; x), \tag{5}$$

where the second equality holds because $\sum_y e^{\text{CLIP}(y, z; x)}$ is a constant of $y$ and exponential function is monotonic. Intuitively, we classify an image by considering the combinations of all possible classes with the ground-truth contextual attributes and identify the class that yields the highest CLIP score.

**Conditioning on ground-truth contextual attributes improves classification accuracy.** We compare the following four methods in zero-shot classification, where the last two are for ablation:

| Conditioning on | Calculation | Prompt example |
|---|---|---|
| No contextual attributes | $\arg\max_y \ \text{CLIP}_1(y; x)$ | *a photo of a {class name of y}.* |
| Ground-truth attribute values | $\arg\max_y \ \text{CLIP}(y, z^*; x)$ | *a photo of a {class name of y}, upside-down.* |
| Wrong attribute values | $\arg\max_y \ \text{CLIP}(y, z_{\text{wrong}}; x)$ | *a photo of a {class name of y}, upright.* |
| Random attribute values | $\arg\max_y \ \text{CLIP}(y, z_{\text{random}}; x)$ | *a photo of a {class name of y}, iaYo5n0Dli7.* |

We evaluate these methods on ImageNet dataset. Similar to Figure 3, we alter easily observable and adjustable attributes such as orientation through image transformations (e.g., vertical flipping). These new attributes become part of the modified images' generation process, for which we have ground-truth annotations. Table 2 shows that compared to not using contextual attributes, conditioning

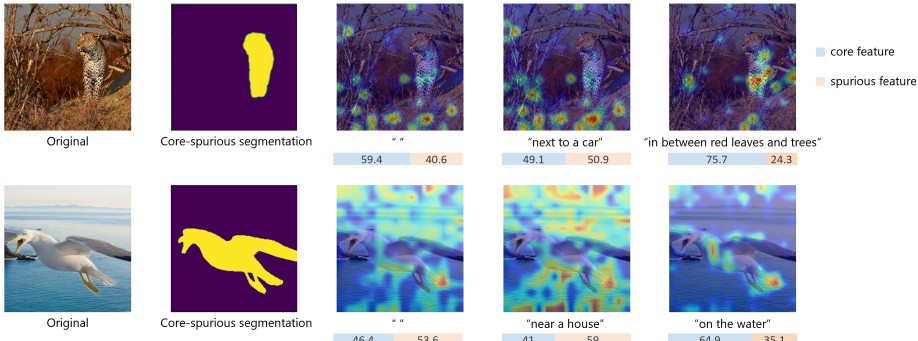

Figure 4: Images of a leopard and a waterbird, core and spurious features, and Grad-CAM heatmaps using no, incorrect, and ground-truth contextual attributes (with text below images). The bar shows core vs. spurious ratio in the heatmap. Visualization shows that classification conditioned on correct contextual attributes enforces CLIP's focus on core features.

on ground-truth contextual attributes improves classification accuracy notably. As an ablation study, conditioning on wrong or randomly generated contextual attributes does not yield similar benefits.

**Conditioning on ground-truth contextual attributes mitigates the reliance on spurious features.**
Contextual attributes like background (e.g., grass) may exhibit spurious correlations with the class (e.g., dog). Classifiers relying on these contextual attributes, also known as spurious features, usually perform poorly. We investigate whether classification conditioned on the known spurious features can enforce CLIP's focus on the object (i.e., core features). As shown in Figure 4, we isolate the background from the core region and employ Grad-CAM (Selvaraju et al., 2017) to identify which region the model focuses on during classification. Specifically, the gradients on pixels with respect to $p(y^*|x, z^*)$, the likelihood of the correct class conditioned on the known background given the image, yields the saliency heatmap. Figure 1 and 4 illustrate that CLIP may rely on spurious features, however, conditioning on correct contextual attributes reduces such reliance and enforces the model to focus on core features, resulting in a more interpretable and reasonable perception (see more results in Appendix E.3). Since image embedding captures both object and background, we suspect that specifying an image's background to CLIP minimizes background influence, potentially sharpening the focus on object features for a better image and text matching in the embedding space.

## 5.2 CONTEXTUAL ATTRIBUTES ARE INFERABLE

The above results highlight the advantages of leveraging CLIP's understanding of contextual attributes. However, manually annotating the attributes is impractical. We now investigate whether CLIP can infer contextual attributes. To infer $z$, we calculate $\arg\max_z \; p(z|x)$ using one of the two approximations in Table 1, where the *ClassAttr* option yields $\arg\max_z \; p(z|x) = \arg\max_z \; \sum_y e^{\texttt{CLIP}(y,z;x)}$, and the *PureAttr* option yields $\arg\max_z \; p(z|x) = \arg\max_z \; \texttt{CLIP}(z;x)$.

**CLIP can infer contextual attributes.** Different from the setting in Section 5.1, we randomly apply transformations to only half of the images in ImageNet. Therefore, inferring each attribute is a binary classification task with a random guessing accuracy of $50\%$. Table 3 shows that the average accuracy is around $74\%$ for both methods, indicating that CLIP can reasonably infer contextual attributes, with some attributes being easier to infer than others. CLIP's understanding of contextual attributes may originate from the numerous captions during the pre-training stage. Moreover, inferring contextual attributes could be easier than determining the object class. Therefore, we may bootstrap CLIP's inference by conditioning on the contextual attributes inferred by itself which is verified in Table 2.

Table 3: Inference accuracy (%) of two contextual attribute inference methods on ImageNet.

| Attribute | vflip | rotation | elastic | invert | solarize | blur | gray | bright | noise | snow | frost | fog | jpeg | Avg |
|---|---|---|---|---|---|---|---|---|---|---|---|---|---|---|
| **ClassAttr** | 76.30 | 68.65 | 72.03 | 78.67 | 74.67 | 62.91 | 84.67 | 56.98 | 66.00 | 86.56 | 82.39 | 89.11 | 66.66 | 74.28 |
| **PureAttr** | 77.31 | 66.01 | 60.00 | 80.61 | 88.79 | 59.26 | 74.26 | 58.94 | 67.16 | 86.56 | 78.23 | 93.95 | 68.71 | 73.83 |

# 6 PerceptionCLIP: EMULATING HUMAN PERCEPTION

Building on the above observations, we propose `PerceptionCLIP`, a two-step zero-shot classification method for CLIP. It emulates the human perception process by first inferring the contextual attributes and then inferring the class conditioning on the contextual attributes. The pseudocode of `PerceptionCLIP` is outlined in Algorithm 1.

---

**Algorithm 1:** `PerceptionCLIP`

**Require :** class $Y$, contextual attributes $\{Z_1, \ldots, Z_m\}$, `CLIP` score (with annotation function $\alpha$), temperature hyperparameter $\tau$

**Input** : image $x$

**Output** : predicted class $\hat{y}$

**Step 1:** infer the distribution of contextual attribute values

$$\hat{p}(z_1, \ldots, z_m | x) \leftarrow \frac{\sum_y e^{\texttt{CLIP}(y, z_1, \ldots, z_m; x)/\tau}}{\sum_y \sum_{z_1, \ldots, z_m} e^{\texttt{CLIP}(y, z_1, \ldots, z_m; x)/\tau}} \text{ or } \frac{e^{\texttt{CLIP}(z_1, \ldots, z_m; x)/\tau}}{\sum_{z_1, \ldots, z_m} e^{\texttt{CLIP}(z_1, \ldots, z_m; x)/\tau}}$$

**Step 2:** infer the class

$$p(y | x, z_1, \ldots, z_m) \leftarrow \frac{e^{\texttt{CLIP}(y, z_1, \ldots, z_m; x)}}{\sum_y e^{\texttt{CLIP}(y, z_1, \ldots, z_m; x)}}$$

$$\hat{y} \leftarrow \arg\max_y \ p(y | x) = \arg\max_y \ \sum_{z_1, \ldots, z_m} p(y | x, z_1, \ldots, z_m) \hat{p}(z_1, \ldots, z_m | x).$$

---

**Step one:** `PerceptionCLIP` estimates the distribution of contextual attributes given an image. Rather than selecting the most probable attribute value, we estimate the entire distribution to accommodate CLIP's inherent uncertainty. In addition, we introduce a temperature hyperparameter $\tau$ to intervene in the estimation. A temperature $\tau$ greater than $1$ smoothens CLIP's estimation, implying less trust in its predictions. The two-step nature also allows for other interventions, such as truncating top $k$ predicted values (i.e., beam search), which we leave for future work.

**Step two:** `PerceptionCLIP` first approximates the class distribution conditioning on each possible contextual attributes' value. Then, it uses the estimated distribution of contextual attributes to calculate the weighted sum of these class distributions, marginalizing out the contextual attributes. Finally, it selects the most probable class $y$ as the predicted output.

**Simplifying into a single step.** It can be seen from Algorithm 1 that setting the temperature to 1 and ignoring constant terms yields $\hat{y} \leftarrow \arg\max_y \sum_{z_1, \ldots, z_m} e^{\texttt{CLIP}(y, z_1, \ldots, z_m; x)}$, essentially simplifying the two-step algorithm into a single step. Intuitively, for each possible class, it sums the exponentiated `CLIP` scores calculated over each contextual attribute value, resulting in an aggregated score for the class. Then, it selects the class with the highest aggregated score.

**Single-step vs. prompt ensembling.** This single-step approach, as a special case of our method, coincides with the prompt ensembling method if we aggregate over some randomly selected attributes (as in 80 templates) instead of all contextual attribute combinations. This coincidence explains the effectiveness of prompt ensembling - it undergoes an implicit perception process. Nevertheless, our experiments show that constructing diverse and systematic prompts using our contextual attribute combinations is superior to ad-hoc template selections in prompt ensembling.

**Two-step vs. single-step.** The one-step method is simpler to implement but lacks two key features. It disallows human intervention when inferring contextual attributes. Our experiments indicate that CLIP does not always infer contextual attributes correctly, whereas human intervention can leverage our prior knowledge to adjust its estimation. Second, the one-step method prevents us from knowing the inferred contextual attributes, which could have improved the interpretability of the results.

**Constructing contextual attributes.** The set of possible contextual attributes is at the core of `PerceptionCLIP`. We construct it with two approaches: 1) We manually construct essential attributes that may be generative factors in the image generation process, especially those causing spurious correlations. This is particularly effective when we know of the dataset. For instance, for the CelebA dataset, we consider gender, age, and race as the attributes. 2) We leverage the in-context learning of large language models for semi-automated construction (shown in Appendix F.4).

Table 4: Zero-shot classification accuracy on five datasets using ViT-B/16. The best result in each column is highlighted in bold, while the next three highest values are underlined.

| Attributes | | ImageNet | ImageNetV2 | ImageNet-R | ImageNet-A | ImageNet-Sketch |
|---|---|---|---|---|---|---|
| single template | | 66.72 | 60.85 | 73.99 | 47.80 | 46.16 |
| 80 templates | | 68.32 | 61.93 | 77.71 | 49.95 | 48.26 |
| | background | 67.98 | 61.65 | 75.87 | 49.85 | 47.08 |
| | illumination | 67.47 | 61.48 | 75.37 | 48.90 | 46.67 |
| | orientation | 67.28 | 61.11 | 74.51 | 48.47 | 46.87 |
| | quality | 68.18 | 61.65 | 76.23 | **50.36** | 47.40 |
| | quantity | 67.64 | 61.46 | 75.37 | 50.04 | 46.59 |
| single attribute | perspective | 67.90 | 61.27 | 75.00 | 49.61 | 46.84 |
| | art | 67.53 | 61.11 | **77.16** | 49.48 | 47.96 |
| | medium | 67.58 | 61.31 | 76.67 | 49.62 | 47.37 |
| | condition | **68.39** | **61.69** | 75.74 | 49.54 | 47.41 |
| | color-scheme | 66.89 | 60.70 | 74.47 | 48.14 | 47.03 |
| | tool | 67.42 | 61.02 | 76.72 | 48.88 | **48.19** |
| composition of top 2 attributes | | 68.52 | 62.28 | 77.78 | 50.88 | 48.46 |
| composition of top 3 attributes | | **68.80** | 62.22 | 78.14 | 51.15 | 48.92 |
| composition of top 4 attributes | | 68.71 | **62.32** | **78.38** | **51.39** | **49.10** |

Table 5: Classification accuracy of ViT-B/16 on different data domains with `PerceptionCLIP`.

| | CUB200 | EuroSAT | Places365 | Flowers102 | Food101 | Oxford Pets |
|---|---|---|---|---|---|---|
| simple template | 56.07 | 51.44 | 38.93 | 67.73 | 88.24 | 88.25 |
| domain template | 56.32 | 54.94 | 38.93 | 70.99 | 88.72 | 89.04 |
| $+ \mathcal{Z}$ | **57.08** | **59.23** | **40.92** | **72.86** | **89.19** | **90.38** |

# 7 EXPERIMENTS

## 7.1 ZERO-SHOT GENERALIZATION

**Settings.** We test `PerceptionCLIP` on ImageNet (Deng et al., 2009) and its out-of-distribution datasets, including ImageNetV2 (Recht et al., 2019), ImageNet-R (Hendrycks et al., 2021a), ImageNet-A (Hendrycks et al., 2021b), and ImageNet-Sketch (Wang et al., 2019). We also test on different data domains (e.g., satellite images), including CUB200 (Wah et al., 2011), EuroSAT (Helber et al., 2019), Places365 (Zhou et al., 2017), Flowers102 (Nilsback & Zisserman, 2008), Food101 (Bossard et al., 2014), and Oxford Pets (Parkhi et al., 2012). For natural images, we compile a set of possible contextual attributes. Each attribute has multiple possible values, and each value has multiple possible descriptions with uniform possibilities to simulate the unknown distribution (details in Appendix E.4). For the dataset in a specific domain, we use domain-specific contextual attributes, for example, *image source* for EuroSAT, *cuisine* for Food101, *species* for Oxford Pets. We use our two-step method (*ClassAttr*) with the temperature as a hyperparameter (details in Appendix E.4).

**Using a single attribute.** Table 4 shows that compared to using the simple template "*a photo of a {class name}*," considering almost any single contextual attribute improves the accuracy, some even surpassing the use of 80 templates. We also observe that the most influential contextual attributes vary for different datasets, potentially attributable to different data generation processes. For example, all images in ImageNet-Sketch are sketches, making *tool* and *art* crucial contextual attributes for image generation. This also indicates that `PerceptionCLIP` works the best when the considered contextual attributes cover the generation process of the dataset.

**Using multiple attributes.** Table 4 also presents the results considering multiple contextual attributes. `PerceptionCLIP`, using the two most effective attributes, can already outperform prompt ensembling using 80 templates across all datasets. As the number of attributes considered increases, the classification accuracy gradually improves. We also test our method on different domains of data in Table 5. The domain templates provided in Radford et al. (2021) already describe the domain in text prompt (e.g., *"a centered satellite photo of {class name}"*) where the domain is a known contextual attribute. As expected, specifying it improves accuracy. `PerceptionCLIP` considers more contextual attributes and further improves zero-shot classification accuracy. For instance, by considering *image source* and *condition* for the EuroSAT dataset, `PerceptionCLIP` achieves a near 5% gain in accuracy.

Ablation studies in Appendix E.4 demonstrate that substituting contextual attributes with random strings markedly reduces performance, highlighting their critical role in our method's effectiveness.

**Intervening in attributes inference.** In Table 6, we evaluate the effectiveness of the intervention. We set temperature $\tau = 3$ and consider the top four attributes. Results show that intervening in inferring contextual attributes achieves modest but consistent performance gains across datasets. In practice, we find that setting the temperature to 3 or 5 usually yields better performance, which also confirms that CLIP cannot perfectly infer contextual attributes. One can also search for the best temperature with a validation set when applicable.

Table 6: Intervening in inferring contextual attributes improves zero-shot classification.

| | Without intervention | With intervention | |
| --- | --- | --- | --- |
| | | ClassAtrr | PureAttr |
| ImageNet | 68.59% | 68.70% | **68.72%** |
| ImageNetV2 | 62.10% | 62.31% | **62.32%** |
| ImageNet-R | 78.12% | **78.38%** | 78.27% |
| ImageNet-A | 51.17% | **51.39%** | 51.22% |
| ImageNet-Sketch | 49.03% | **49.10%** | **49.10%** |

## 7.2 GROUP ROBUSTNESS

Group robustness is a critical measure of a model's bias. It measures the ability to perform consistently across different subgroups within a dataset (Liu et al., 2021). We evaluate the group robustness of `PerceptionCLIP` through bird type classification on the Waterbirds dataset (Sagawa et al., 2020) and hair color classification on the CelebA (Liu et al., 2015) dataset. In both datasets, each image has an underlying group attribute unknown to the model. These group attributes are *background* in Waterbirds and *gender* in CelebA. They both spuriously correlate with the class but do not causally determine the class. To evaluate the worst-group accuracy, we group images by their classes and attributes, then assess each group's accuracy following Sagawa et al. (2020). Table 7 and 8 show that when the text prompts only describe the class and ignore contextual attributes (first row), such as "*a photo of a {landbird/waterbird}*" and "*a photo of a celebrity with {dark hair/blond hair}*," CLIP exhibits biased accuracy, with a significant discrepancy between average accuracy and the worst-group accuracy. This bias arises because CLIP overly relies on spurious features, such as associating images with a water background to the waterbird class, instead of focusing on the bird. As shown in Figure 4, conditioning on group attributes such as background helps reduce CLIP's reliance on spurious features, making the model less biased. Results in Table 7 and 8 also confirm that by considering *background* (with values in {*on land*, *on water*}) for Waterbird dataset, and *gender* (with values in {*female*, *male*}) for CelebA dataset, `PerceptionCLIP` reduces the accuracy gap in most cases. By incorporating more values (e.g., *in forest*) into the attribute *background*$^+$, or considering more contextual attributes like *age* and *race*, the group robustness can be further improved.

Table 7: Average accuracy and worst group accuracy on the Waterbirds dataset.

| | RN50 | | | ViT-B/32 | | | ViT-B/16 | | | ViT-L/14 | | |
| --- | --- | --- | --- | --- | --- | --- | --- | --- | --- | --- | --- | --- |
| | Avg ↑ | Worst ↑ | Gap↓ | Avg ↑ | Worst ↑ | Gap ↓ | Avg ↑ | Worst ↑ | Gap ↓ | Avg ↑ | Worst ↑ | Gap ↓ |
| without $\mathcal{Z}$ | 90.47 | 16.07 | 74.40 | 87.34 | 47.28 | 40.06 | 87.34 | 26.79 | 60.56 | 90.55 | 44.64 | 45.91 |
| $\mathcal{Z}$={background} | 88.78 | 16.07 | 72.71 | 89.80 | 66.07 | 23.73 | 82.98 | 16.07 | 66.91 | 86.44 | 44.94 | 41.51 |
| $\mathcal{Z}$={background$^+$} | 90.32 | 35.71 | **54.61** | 78.60 | 60.33 | **18.28** | 85.80 | 41.07 | **44.73** | 87.74 | 61.12 | **26.62** |

Table 8: Average accuracy and worst group accuracy on the CelebA dataset.

| | RN50 | | | ViT-B/32 | | | ViT-B/16 | | | ViT-L/14 | | |
| --- | --- | --- | --- | --- | --- | --- | --- | --- | --- | --- | --- | --- |
| | Avg ↑ | Worst ↑ | Gap↓ | Avg ↑ | Worst ↑ | Gap ↓ | Avg ↑ | Worst ↑ | Gap ↓ | Avg ↑ | Worst ↑ | Gap ↓ |
| without $\mathcal{Z}$ | 81.05 | 73.87 | 7.19 | 80.73 | 75.82 | 4.91 | 75.16 | 62.01 | 13.16 | 86.98 | 77.36 | 9.61 |
| $\mathcal{Z}$={gender} | 85.10 | 80.44 | 4.65 | 79.89 | 76.70 | **3.19** | 75.27 | 65.13 | 10.14 | 80.30 | 74.31 | 5.99 |
| $\mathcal{Z}$={gender, age} | 87.71 | 84.98 | **2.74** | 82.82 | 78.06 | 4.76 | 75.81 | 65.52 | 10.29 | 82.26 | 79.06 | 3.21 |
| $\mathcal{Z}$={gender, age, race} | 85.55 | 82.51 | 3.05 | 82.02 | 75.94 | 6.09 | 77.17 | 69.18 | **7.99** | 83.04 | 80.84 | **2.20** |

## 8 CONCLUSION

This paper proposes `PerceptionCLIP`, a zero-shot classification method for CLIP that emulates human visual perception. By doing classification conditioned on self-inferred contextual attributes, it achieves improved generalization, less reliance on spurious features, and improved group robustness. One limitation of our method is its sensitivity to text descriptions. Although using a distribution of descriptions alleviates this sensitivity, it is an intrinsic problem of CLIP itself. Future work may overcome this limitation by using advanced vision-language models. Another future direction is applying this technique to pre-training and fine-tuning stages (see more in Appendix G).

ACKNOWLEDGEMENT

An, Zhu, Panaitescu-Liess and Huang are supported by National Science Foundation NSF-IIS-2147276 FAI, DOD-ONR-Office of Naval Research under award number N00014-22-1-2335, DOD-AFOSR-Air Force Office of Scientific Research under award number FA9550-23-1-0048, DOD-DARPA-Defense Advanced Research Projects Agency Guaranteeing AI Robustness against Deception (GARD) HR00112020007, Adobe, Capital One and JP Morgan faculty fellowships.

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

# APPENDIX

## A   EXTENDED RELATED WORK

**Descriptive prompts with external knowledge.** Due to CLIP's ability to understand finer-grained visual concepts beyond classes (e.g., body parts and components), some work leverages external knowledge to augment prompts with additional visual concepts to improve CLIP's zero-shot classification. For example, Menon & Vondrick (2022); Pratt et al. (2022); Mao et al. (2022); Feng et al. (2023) use large language models (LLMs) like GPT-3 to generate class-specific descriptions for each class and incorporate them into prompts, resulting in prompts like "a photo of a hen, which has two legs". Novack et al. (2023) use class hierarchies (existing or by querying GPT-3) to generate sub-classes for each parent class and aggregate model predictions on all sub-classes to get a final prediction. Udandarao et al. (2023) use class names to retrieve and maintain some auxiliary data to help downstream classification. In contrast, our method addresses class-independent attributes (i.e., contextual attributes) such as background and orientation, whose comprehension by CLIP is not well-known. These attributes are also combinatorial, potentially covering more aspects of an image than class-specific attributes. Moreover, we can still leverage contextual attributes (e.g., gender, age) when class-specific attributes are hard to articulate, as in the hair-color classification tasks on CelebA. We also find that specifying spurious contextual attributes reduces distractions from their spurious correlations.

**Does CLIP truly understand descriptive prompts?**  Some work investigates a seemingly obvious question: do these descriptive prompts play a role in CLIP's prediction? Roth et al. (2023) show that replacing class-specific descriptions in prior work with random words or even meaningless characters can achieve similar performance, resembling the effect of noise augmentation or randomized smoothing. Li et al. (2023c) find that GLIP (a similar VLM as CLIP), often disregards contextual information in the prompts and relies heavily on class names in object detection. Addressing these findings, we ablate our method and show that random attributes or meaningless characters yield approximately half the benefit compared to using correct or self-inferred attributes, indicating that our method's effectiveness stems from the proper use of contextual attributes instead of noise augmentation. Roth et al. (2023) also show that appending high-level class-independent descriptions (e.g., "food" for Food101, "place" for Places365) to prompts helps classification, which aligns with our findings.

**Prompt tuning.** Another line of work that modifies prompts to improve CLIP's classification is prompt tuning, which optimizes the prefix characters of the prompts. Typical prompt tuning methods require labeled (Zhou et al., 2022b;a; Zhu et al., 2022; Derakhshani et al., 2023) or unlabeled downstream data (Huang et al., 2022; Mirza et al., 2023; Menghini et al., 2023), making them fall outside our scope of zero-shot (data-free) classification. They are also prone to overfitting the training dataset, whereas our method relies on general image attributes (e.g, illumination) shared by common datasets. On the other hand, Shu et al. (2022) use test-time prompt tuning that applies to zero-shot classification. Specifically, they generate multiple views for each test image and optimize the prompt to minimize the entropy of the model's prediction on these views. This method introduces several hyperparameters that require tuning on a labeled proxy validation set. In contrast, our method, depending on implementation, introduces either no additional hyperparameters or only one (temperature). Furthermore, our method is training-free and can work in the black-box setting.

**Reasoning and chain-of-thoughts.** The inference process of our method resembles the reasoning or chain-of-thoughts in prompting LLMs (Wei et al., 2022; Yao et al., 2023), where the model is prompted to give some intermediate step results and then conditioning on them to give final results. However, CLIP itself cannot do step-wise reasoning out of the box, so our method manually prompts it through the reasoning process.

## B   IMAGE CAPTION EXAMPLES

In the pertaining stage, the human-written caption for each image typically describes the visual object, encompassing its class and a few contextual attributes. We show some caption examples in Table 9, chosen from a similar dataset LAION-400M (Schuhmann et al., 2021), since the original pretraining dataset of CLIP is not made public. We can see that those captions not only describe class but also contextual attributes like color, style, and background.

Table 9: Image caption examples from LAION-400M (comparable to CLIP's pretraining dataset).

| | |
|---|---|
| Caption #1 | *Men's Classics Round Bracelets Watch in Grey* |
| Caption #2 | *stock photo of gremlins - 3 d cartoon cute green gremlin monster - JPG* |
| Caption #3 | *Medium Size of Chair: Fabulous Mid Century Modern Chair Adalyn Accent In Red:* |

## C  HUMAN VISUAL PERCEPTION

The classic neuroscience textbook Kandel et al. (2013) offers a modern view of human visual perception, presenting a significant difference from current zero-shot classification methods:

> *"The brain analyzes a visual scene at three levels: low, intermediate, and high. At the lowest level, visual attributes such as local contrast, orientation, color, and movement are discriminated. The intermediate level involves analysis of the layout of scenes and of surface properties, parsing the visual image into surfaces and global contours, and distinguishing foreground from background. The highest level involves object recognition."*
>
> *"... the perceptual interpretation we make of any visual object depends not just on the properties of the stimulus but also on its context, on other features in the visual field."*

This perception process is hierarchical, cascaded, and context-dependent, differing from current zero-shot classification methods, which overlook contextual attributes. In this paper, we propose `PerceptionCLIP` to mimic human perception process.

## D  APPROXIMATING CONDITIONAL PROBABILITIES

With the energy-function-like `CLIP` score

$$\text{CLIP}(y, z_1, \ldots, z_m; x) \triangleq \left\langle \phi_I(x), \frac{\mathbb{E}\ \phi_T\big(\alpha(y) \oplus \alpha(z_1) \oplus \cdots \oplus \alpha(z_m)\big)}{\|\mathbb{E}\ \phi_T\big(\alpha(y) \oplus \alpha(z_1) \oplus \cdots \oplus \alpha(z_m)\big)\|} \right\rangle, \quad (6)$$

we first approximate the joint conditional probability $p(y, z_1, \ldots, z_m | x)$. It measures the likelihood of an object class and some contextual attributes occurring together given the image as

$$p(y, z_1, \ldots, z_m | x) \triangleq \frac{e^{\text{CLIP}(y, z_1, \ldots, z_m; x)}}{\sum_y \sum_z e^{\text{CLIP}(y, z_1, \ldots, z_m; x)}} \quad (7)$$

which is essentially the normalization of the exponential of the `CLIP` score. Then, we derive the conditional probability $p(y | z_1, \ldots, z_m, x)$, which measures the probability of an object class given both the image and the contextual attributes as

$$p(y | x, z_1, \ldots, z_m) = \frac{p(y, z_1, \ldots, z_m | x)}{p(z_1, \ldots, z_m | x)} \quad (8)$$

$$= \frac{p(y, z_1, \ldots, z_m | x)}{\sum_y p(y, z_1, \ldots, z_m | x)} \quad (9)$$

$$= \frac{e^{\text{CLIP}(y, z_1, \ldots, z_m; x)}}{\sum_y e^{\text{CLIP}(y, z_1, \ldots, z_m; x)}} \quad (10)$$

using the definition of joint probability and the rules of conditional probability. Next, we approximate the conditional probability $p(z_1, \ldots, z_m | x)$, which measures the likelihood of some contextual attributes given the image as

$$p(z_1, \ldots, z_m | x) = \sum_y p(y, z_1, \ldots, z_m | x) \quad (11)$$

$$= \frac{\sum_y e^{\text{CLIP}(y, z_1, \ldots, z_m; x)}}{\sum_z \sum_y e^{\text{CLIP}(y, z_1, \ldots, z_m; x)}}. \quad (12)$$

It sums up the probabilities of contextual attributes appearing in each class to give a total probability of them appearing in the image. We named this method *ClassAttr*. Another simplified way, named *PureAttr*, ignores the classes and use

$$p(z_1, \ldots, z_m | x) \approx \frac{e^{\mathtt{CLIP}(z_1, \ldots, z_m; x)}}{\sum_z e^{\mathtt{CLIP}(z_1, \ldots, z_m; x)}} \tag{13}$$

to do the estimation. Here, we only consider the contextual attributes in the `CLIP` score with descriptions like "*a photo of an object, {description of z}*" where we use a word like "object" instead of a particular class, making the `CLIP` score class-agnostic. In our experiments, the first version occasionally outperformed the second, although the performance of the two is generally similar.

# E    EXPERIMENTAL DETAILS

Table 10: Summary of descriptions for different attributes used in Figure 3, Table 2 and Table 3. $z^*$ denotes the correct value of the contextual attribute, and $z_{wrong}$ denotes the wrong value of the contextual attribute. Ideally, each attribute has a distribution of text descriptions. Here, we use three descriptions and use the averaged text embeddings of them to calculate the CLIP score.

| Attribute | $\alpha(y) \oplus \alpha(z^*)$ | $\alpha(y) \oplus \alpha(z_{wrong})$ |
|---|---|---|
| vertical flip | "a photo of a {y}." 
 "a photo of a {y}, upside-down." 
 "a photo of a {y}, the photo is upside-down." | "a photo of a {y}." 
 "a photo of a {y}, upright." 
 "a photo of a {y}, the photo is upright." |
| 90° rotation | "a photo of a {y}." 
 "a photo of a {y}, rotated." 
 "a photo of a {y}, the photo is rotated." | "a photo of a {y}." 
 "a photo of a {y}, upright." 
 "a photo of a {y}, the photo is upright." |
| elastic-transform | "a photo of a {y}." 
 "a photo of a {y}, with distortion." 
 "a photo of a {y}, the photo is distorted." | "a photo of a {y}." 
 "a photo of a {y}, normal." 
 "a photo of a {y}, the photo is normal." |
| color-invert | "a photo of a {y}." 
 "a photo of a {y}, color-inverted." 
 "a photo of a {y}, the photo is color-inverted." | "a photo of a {y}." 
 "a photo of a {y}, normal." 
 "a photo of a {y}, the photo is normal." |
| solarize | "a photo of a {y}." 
 "a photo of a {y}, solarized." 
 "a photo of a {y}, the photo is solarized." | "a photo of a {y}." 
 "a photo of a {y}, normal." 
 "a photo of a {y}, the photo is normal." |
| blur | "a photo of a {y}." 
 "a photo of a {y}, blurred." 
 "a photo of a {y}, the photo is blurred." | "a photo of a {y}." 
 "a photo of a {y}, clear." 
 "a photo of a {y}, the photo is clear." |
| grayscale | "a photo of a {y}." 
 "a photo of a {y}, grayscale." 
 "a photo of a {y}, the photo is in black and white." | "a photo of a {y}." 
 "a photo of a {y}, colorful." 
 "a photo of a {y}, the photo is colorful." |
| bright | "a photo of a {y}." 
 "a photo of a {y}, bright." 
 "a photo of a {y}, the photo is bright." | "a photo of a {y}." 
 "a photo of a {y}, dark." 
 "a photo of a {y}, the photo is dark." |
| noise | "a photo of a {y}." 
 "a photo of a {y}, with noise." 
 "a photo of a {y}, the photo has noise." | "a photo of a {y}." 
 "a photo of a {y}, clear." 
 "a photo of a {y}, the photo is clear." |
| snow | "a photo of a {y}." 
 "a photo of a {y}, in the snow." 
 "a photo of a {y}, the photo is in the snow." | "a photo of a {y}." 
 "a photo of a {y}, clear." 
 "a photo of a {y}, the photo is clear." |
| frost | "a photo of a {y}." 
 "a photo of a {y}, in the frost." 
 "a photo of a {y}, the photo is in the frost." | "a photo of a {y}." 
 "a photo of a {y}, clear." 
 "a photo of a {y}, the photo is clear." |
| fog | "a photo of a {y}." 
 "a photo of a {y}, in the fog." 
 "a photo of a {y}, the photo is in the fog." | "a photo of a {y}." 
 "a photo of a {y}, clear." 
 "a photo of a {y}, the photo is clear." |
| jpeg | "a photo of a {y}." 
 "a photo of a {y}, in jpeg format." 
 "a photo of a {y}, the photo is in jpeg format." | "a photo of a {y}." 
 "a photo of a {y}, in high resolution." 
 "a photo of a {y}, the photo is in high resolution." |

## E.1    DETAILS ON THE EVALUATION IN FIGURE 3 AND TABLE 2

We do evaluation on the ImageNet dataset. Due to the lack of annotated contextual attributes, we consider some easily observable and adjustable attributes, including image orientation, illumination, etc. We first examine and confirm that most ImageNet images share the same attribute values, including upright orientation, natural illumination, and standard image quality. However, these default values are too trivial, making their textual descriptions unlikely to appear in the captions of the

Table 11: Similarity score and classification accuracy on ImageNet test set. We apply a composition of two transformation functions on images, and use the composition of attributes' descriptions for text.

| Attributes | Similarity (CLIP score $\times$ 100) | | | |
|---|---|---|---|---|
| | w/o $z$ | w/ random $z$ | w/ wrong $z$ | w/ correct $z$ |
| vertical flip + color-invert | 25.39 | 28.23 (↑2.84) | 26.28 (↑0.88) | 30.26 (↑**4.86**) |
| grayscale + elastic-transform | 26.66 | 30.55 (↑3.89) | 26.48 (↓0.19) | 32.15 (↑**5.49**) |
| **Attributes** | Accuracy (%) | | | |
| | w/o $z$ | w/ random $z$ | w/ wrong $z$ | w/ correct $z$ |
| vertical flip + color-invert | 19.44 | 20.88 (↑1.44) | 20.01 (↑0.57) | 21.32 (↑**1.88**) |
| grayscale + elastic-transform | 29.79 | 30.49 (↑0.70) | 30.14 (↑0.35) | 30.59 (↑**0.80**) |

pretraining data. Therefore, we then alter these attribute values through certain image transformations (e.g., vertical flipping), thus making the new attribute values have non-trivial descriptions. These new attribute values become part of the modified images' data generation process, for which we have ground-truth annotations.

**Contextual attributes and their descriptions.** We separately apply thirteen image transformation functions (e.g., vertical flip) to all the ImageNet test images. Note that the last five transformations (i.e., noise, snow, frost, fog, jpeg) are tested directly on the ImageNet-C dataset (Hendrycks & Dietterich, 2018), which contains the same images as in the ImageNet test set while images are corrupted with certain transformations. We use relatively strong strengths in those transformations, ensuring nonnegligible generative factors. This is also why CLIP has degraded performance on these corrupted data. Table 10 shows the descriptions we used in this evaluation. When we ignore the contextual attribute, we use a simple template, "*a photo of a {class name}.*". When considering the contextual attribute, we test the cases using the correct attribute value (e.g., upside-down) with "*a photo of a {class name}, upside-down.*" and the wrong attribute value (e.g., upright) with "*a photo of a {class name}, upright.*", respectively. For each contextual value, we use three descriptions to simulate the distribution of descriptions and average their embeddings as the text embedding to calculate the CLIP score.

**Randomized descriptions.** To validate the effectiveness of contextual attributes, we also compare with the cases where we use random descriptions as proposed in Roth et al. (2023). We replace every word in $\alpha(z^*)$ with random characters while keeping the word length unchanged. For example, $\alpha(y) \oplus \alpha(z_{random})$ of vertical flip contains three descriptions: "*a photo of a {y}.*", "*a photo of a {y}, iaYo5n0Dli7.*", "*a photo of a {y}, 8g2, Me5tx, q1, 6Ud2ole94Ir.*"

**Additional results on the composition of contextual attributes.** We perform the same evaluation while considering more than one contextual attribute. Table 11 shows the results. We draw the same conclusion that correct contextual attribute values lead to better image-text alignment and higher classification accuracy.

**Ablation on the increased similarities.** The CLIP score measures the similarity between the image and the text prompt. In Figure 3, we observe that correctly matched image and attribute pairs provide higher CLIP scores for the ground-truth class than mismatched pairs using wrong or random contextual attributes. It indicates that incorporating correct contextual attributes yields the greatest benefit for the ground-truth class. Additionally, we delve into the impact of including these correct attributes on both the correct and incorrect classes. We calculate the increase in CLIP scores for class $y$ with $\Delta\texttt{CLIP}(y) \triangleq \texttt{CLIP}(y, z^*; x) - \texttt{CLIP}_1(y; x)$, and the increase in prediction probabilities for class $y$ with $\Delta p(y) \triangleq p(y|x, z^*) - p(y|x)$.

In Figure 5 (left), we compare the $\Delta\texttt{CLIP}(y^*)$ and $\Delta\texttt{CLIP}(y_{wrong})$, where the latter one is the average increase of the Top-K wrong classes. As expected, incorporating ground-truth attributes into text prompts results in increased scores for both correct and incorrect classes, and the correct class benefits more from this enhancement, as the accurate description of the class and the attribute, achieves a better alignment with the corresponding image. Figure 3 and 5 together validate that the

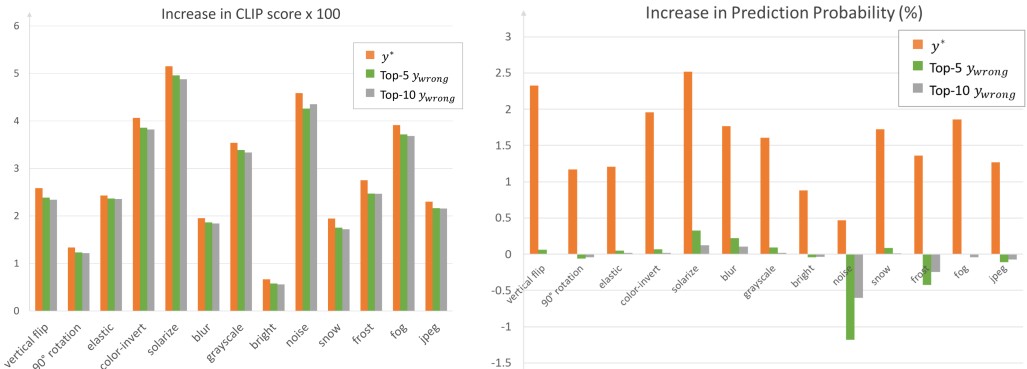

Figure 5: The increase in (left) `CLIP` scores and the (right) prediction probabilities by incorporating the descriptions of the correct contextual attribute into the text prompts. We compare the increased `CLIP` scores and prediction probabilities for the ground-truth class $y^*$, the Top-5 and Top-10 wrong classes. (left) Incorporating ground-truth attributes into text prompts results in increased `CLIP` scores for both correct and incorrect classes. This improvement is attributed to the enhanced alignment of the text prompts with the images, addressing previously overlooked contextual attributes. Notably, the `CLIP` score of the correct class benefits more from this enhancement for all the attributes considered. This is because the accurate description of the class, combined with the contextual attributes, achieves a more precise alignment with the corresponding image. (right) Therefore, the model is more likely to predict the correct class after being provided with the correct context description in the prompt.

CLIP model understands the contextual attributes, and describing correct class and attributes yields higher similarity scores as described in Equation 4.

In Figure 5 (right), we further compare the increase in the prediction probabilities for $\Delta p(y)$ and $\Delta p(y_{wrong})$ where the probability is calculated by applying softmax on CLIP scores and is used for the final classification. By incorporating correct context, the prediction probability of the correct class increased significantly, while the wrong classes got unchanged or even decreased probabilities. The predicted probability for the correct class increases by an average of 1.5%. The predicted probabilities for top-5 and top-10 wrong classes decrease by an average of 0.07% and 0.05%. Such findings also explain the increased accuracy in Table 2 when incorporating the correct contextual attributes.

### E.2 DETAILS ON THE EVALUATION IN TABLE 3

In Table 3, we test whether CLIP can infer underlying contextual attributes by itself. In this experiment, we only apply transformations to half of the images from the ImageNet test set and use descriptions shown in Table 10. The task is to predict the correct contextual attribute's value, which is a binary classification task. For example, half images are upright, and half images are upside-down, and the goal is to classify the orientation of the images by CLIP. We evaluate two methods with two approximations of $p(z|x)$ in Table 1. Note that we do not intervene in the inference in this experiment.

### E.3 DETAILS ON THE VISUALIZATION

We consider some spatially separable spurious attributes (also known as spurious features), such as backgrounds, and annotate core regions and spurious regions with the help of *Segment Anything* (Kirillov et al., 2023). Then, we use Grad-CAM to generate a heatmap for salient pixels. first, we use a function that computes CLIP's similarity score for each class `CLIP`$(y, z^*; x)$ and apply softmax on top of these values. Then, we only consider the scalar value corresponding to the ground-truth class which is essentially the conditional probability $p(y^*|x, z^*)$. We compute the gradients using the layer before the final attention block of ViT-L/14 as suggested in a popular explainability library.[1] Intuitively, the regions where the salient pixels are located are the regions the model pays attention

---

[1]https://github.com/jacobgil/pytorch-grad-cam

Table 13: Summary of contextual attributes and their value descriptions used in ImageNet-related datasets.

| Attributes | Values |
|---|---|
| *orientation* | upright, upside-down, rotated |
| *background* | others, natural, urban, indoor |
| *quality* | normal, good, bad, low res, pixelated, jpeg corrupted, blurry, clean, dirty |
| *illumination* | normal, bright, dark |
| *quantity* | others, many, one, large, small |
| *perspective* | normal, close up, cropped, obscured |
| *art* | non-art, others, sculpture, rendering, graffiti, tattoo, embroidery, paper art, sketch, cartoon |
| *medium* | others, video game, plastic, toy |
| *condition* | normal, cool, nice, weird |
| *color-scheme* | normal, black and white |
| *tool* | others, pencil, pen, digital tool |

to when making predictions, and we hope that the model focuses as much as possible on regions of core features (i.e., features with causal correlation to classes). Note that, adding a spurious attribute's description in this evaluation won't really make the model look at it when classifying because all the descriptions (for all classes) will contain that attribute.

To compute the ratio between the usage of core and spurious regions in prediction, we: *(1)* run Segment Anything (Kirillov et al., 2023) and select a segmentation mask for the core region of each image (e.g., the bird or the leopard), then consider the rest (non-core) regions as spurious; *(2)* use the masks to identify the core and spurious pixel values from the Grad-CAMs and compute the mean pixel value for both of these regions; *(3)* normalize the numbers and show them as two percentages in a bar plot for each Grad-CAM.

In Table 12, we quantitatively evaluate the model reliance on core feature and spurious feature. We use ViT-B/32 as the image encoder and evaluate it on the Waterbirds test set. The dataset contains images of land birds and water birds on land or on the water. We use the same method introduced above to calculate the ratio of core versus spurious through Grad-CAM. We use "on land" and "on water" to describe the context (e.g., background). We compare the correct context with no context, wrong context, and random context where the random context is

Table 12: The average saliency (%) of the core feature and the spurious feature evaluated on the Waterbirds test set.

| | **Core** (↑) | **Spurious** (↓) |
|---|---|---|
| no context | 62.8 | 37.2 |
| wrong context | 62.6 | 37.4 |
| random context | 62.3 | 37.7 |
| correct context | **66.3** | **33.7** |

the random string that replaces the correct context while keeping the description length unchanged. Results in Table 12 also indicate that, by incorporating the correct context, the model relies more on the core feature when doing the classification.

### E.4 DETAILS ON THE EXPERIMENTS IN SECTION 7

In Table 4, we test `PerceptionCLIP` on ImageNet and its OOD datasets. We first use GPT-4 to summarize the contextual attributes involved in the 80 hand-crafted templates (Radford et al., 2021), then add three contextual attributes (orientation, background, tool) to the testing bed. Table 13 shows the values of every contextual attribute. We use multiple descriptions to describe every attribute value, and use their average text embedding of the full sentence in the implementation. When considering a single attribute, we use a main template, "*a photo of a {class name}*" and concatenate it with the description of each attribute value. When considering the composition of attributes, we generate combinations from the values across all attributes. Such simple concatenation of descriptions works well, probably because the pre-trained CLIP model behaves like a bag-of-words (Yuksekgonul et al., 2023). Future works could explore better ways of composing text prompts.

Table 14 and 15 list the contextual attributes used in Table 5, 7 and 8. Attributes and their values are manually designed based on our priors of datasets. Experiments on Waterbirds and CelebA are conducted on their training set.

Table 14: Datasets, domain templates and contextual attributes used in Table 5

| Dataset | Domain Template | Attributes |
|---------|-----------------|------------|
| CUB200 | "a photo of a $\{y\}$, a type of bird" | *size, background, condition* |
| EuroSAT | "a centered satellite photo of $\{y\}$" | *condition, source* |
| Places365 | "a photo of a $\{y\}$" | *background, quality, condition* |
| Flowers102 | "a photo of a $\{y\}$, a type of flower" | *background, illumination, quality, condition* |
| Food101 | "a photo of a $\{y\}$, a type of food" | *cuisines, condition* |
| Oxford Pets | "a photo of a $\{y\}$, a type of pet" | *species, background, pose, interaction* |

Table 15: Domain templates, contextual attributes and their descriptions used in Table 7 and Table 8

| Dataset | Domain Template | Attributes | Values |
|---------|-----------------|------------|--------|
| Waterbirds | "a photo of a $\{y\}$" | *background* | on land, on water |
| | | *background$^+$* | + in forest, in sky, on street, on grass, on tree, with flowers, on beach, with human, on a branch |
| CelebA | "a photo of a celebrity with $\{y\}$" | *gender* | female, male |
| | | *age* | young, old |
| | | *race* | white skin, dark skin, asian |

All the above experiments use *ClassAttr* version of `PerceptionCLIP` and the intervention by setting a temperature $\tau$ in the first step (i.e., inferring contextual attributes). We found that mildly smoothing the estimation by setting $\tau$ to be 3 or 5 usually has the best performance. When we do not have a good prior of the temperature, just setting it to 1 can also have relatively good results. The reported numbers in our experiments use a temperature selected from {1,3,5,10} that performs the best on the particular dataset.

**Ablation studies.** In Table 4 and 5, by incorporating contextual attributes, `PerceptionCLIP` improves the zero-shot classification accuracy in all cases. Adding descriptions of contextual attributes to text prompts has two effects: 1) it introduces more tokens to the text prompt, 2) and the tokens describe the contextual attributes. To figure out which effect causes the improvement, we conduct ablation studies by replacing the descriptions of contextual attributes with the same-length random strings. In Table 16, we do ablation studies on the best attribute composition for all datasets. In Table 17, we keep the domain template but randomize other contextual attributes. For every dataset, we run 5 times with random seeds and report the mean and variance. Results show that adding more tokens can improve the accuracy marginally in most cases, but can also decrease the accuracy as in the case of EuroSAT and Oxford Pets. The improvement brought by adding random tokens might be a result of augmentation (Jain et al., 2023) or register (Darcet et al., 2023). More importantly, there is a significant performance gap between using the random strings and the descriptions of contextual attributes, suggesting that the improvement provided by our method primarily stems from the incorporation of contextual attributes.

Table 16: Ablation study on ImageNet and related datasets.

| Attributes | ImageNet | ImageNetV2 | ImageNet-R | ImageNet-A | ImageNet-Sketch |
|------------|----------|------------|------------|------------|-----------------|
| CLIP | 66.72 | 60.85 | 73.99 | 47.80 | 46.16 |
| PerceptionCLIP | **68.80** | **62.32** | **78.38** | **51.39** | **49.10** |
| ablation w/ random | $67.59 \pm 0.27$ | $61.27 \pm 0.11$ | $75.53 \pm 0.28$ | $49.74 \pm 0.37$ | $47.63 \pm 0.23$ |

# F  ADDITIONAL RESULTS AND ANALYSIS

## F.1  COMPUTATIONAL COMPLEXITY.

Similar to the implementation of prompt ensembling, we pre-compute the embeddings of all class and contextual attribute combinations, and then use these pre-computed embeddings in each inference process. Since we use the average of text embeddings when there are multiple descriptions for

Table 17: Ablation study on different data domains.

| | CUB200 | EuroSAT | Places365 | Flowers102 | Food101 | Oxford Pets |
|---|---|---|---|---|---|---|
| domain template | 56.32 | 54.94 | 38.93 | 70.99 | 88.72 | 89.04 |
| + $\mathcal{Z}$ | **57.08** | **59.23** | **40.92** | **72.86** | **89.19** | **90.38** |
| + random | $56.68 \pm 0.17$ | $53.58 \pm 2.34$ | $39.98 \pm 0.37$ | $71.41 \pm 0.45$ | $88.89 \pm 0.08$ | $88.35 \pm 0.45$ |

Table 18: Performance of `PerceptionCLIP` using two order types in the attribute concatenation.

| Order | ImageNet | ImageNetV2 | ImageNet-R | ImageNet-A | ImageNet-Sketch |
|---|---|---|---|---|---|
| forward | **68.71** | **62.32** | 78.25 | **51.39** | 48.97 |
| backward | 68.71 | 62.15 | **78.38** | 51.21 | **49.10** |

one value, our method needs multiple forward passes to get the text embeddings, causing a longer preparation time. Since these computations are one-time, the time complexity during inference is unaffected by the number of contextual attributes. Compared to the basic method, which stores $O(|\mathcal{Y}|)$ embedding vectors, this implementation needs to store $O(|\mathcal{Y}| \times |\mathcal{Z}_1| \times \cdots \times |\mathcal{Z}_m|)$ embedding vectors. The space complexity limits the number of contextual attributes considered in practice. We will consider using beam search to only reserve top-k attributes to reduce the space storage requirement in future work.

## F.2 AN ANALYSIS OF THE ORDER IN ATTRIBUTE COMBINATION

When considering multiple contextual attributes, we concatenate their textual descriptions. An interesting question is whether their order in the text affects the performance. In Table 18, we test two order types when combining the top four attributes in the ImageNet experiments. The forward direction ordering attributes from the most powerful to the last. The backward direction does the opposite ordering. Unfortunately, we do not observe a good way of ordering consistently outperforming others. We suspect that it is due to CLIP's sensitivity to the text, and the direct concatenation may not be the best way of combining attributes to approximate the distributions of captions in the pertaining stage.

## F.3 MORE VISUALIZATIONS

We show more visualizations in Figure 6 and 7. Figure 6 shows images from the ImageNet dataset with the ground-truth class *leopard*. Figure 7 shows images from the Waterbirds dataset with the ground-truth class *waterbird*. Grad-CAMs show that CLIP relies more on core features when conditioned on the correct contextual attributes (e.g., background) for classification. The reliance on core features also improves model interpretability.

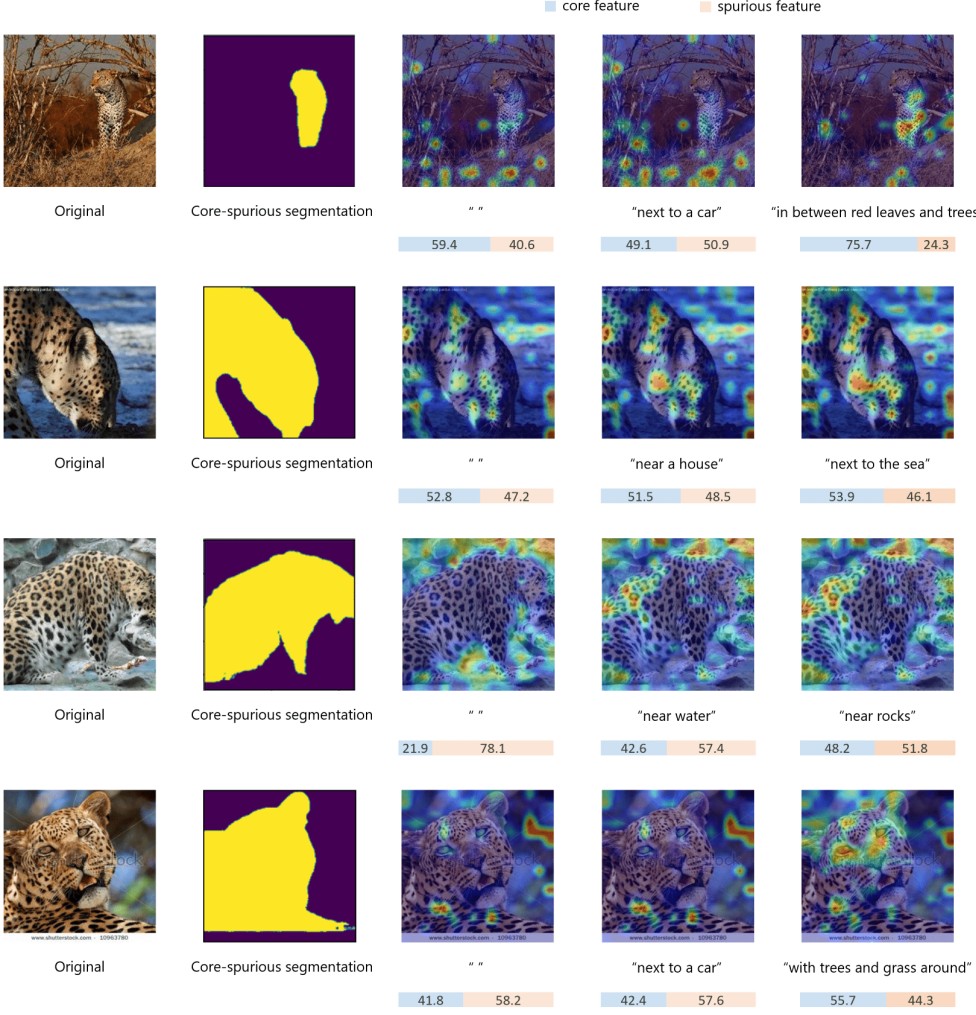

Figure 6: Leopard images from ImageNet dataset. Visualization of the original image, the regions of core and spurious features, and the Grad-CAMs obtained using no, incorrect, and ground-truth contextual attributes.

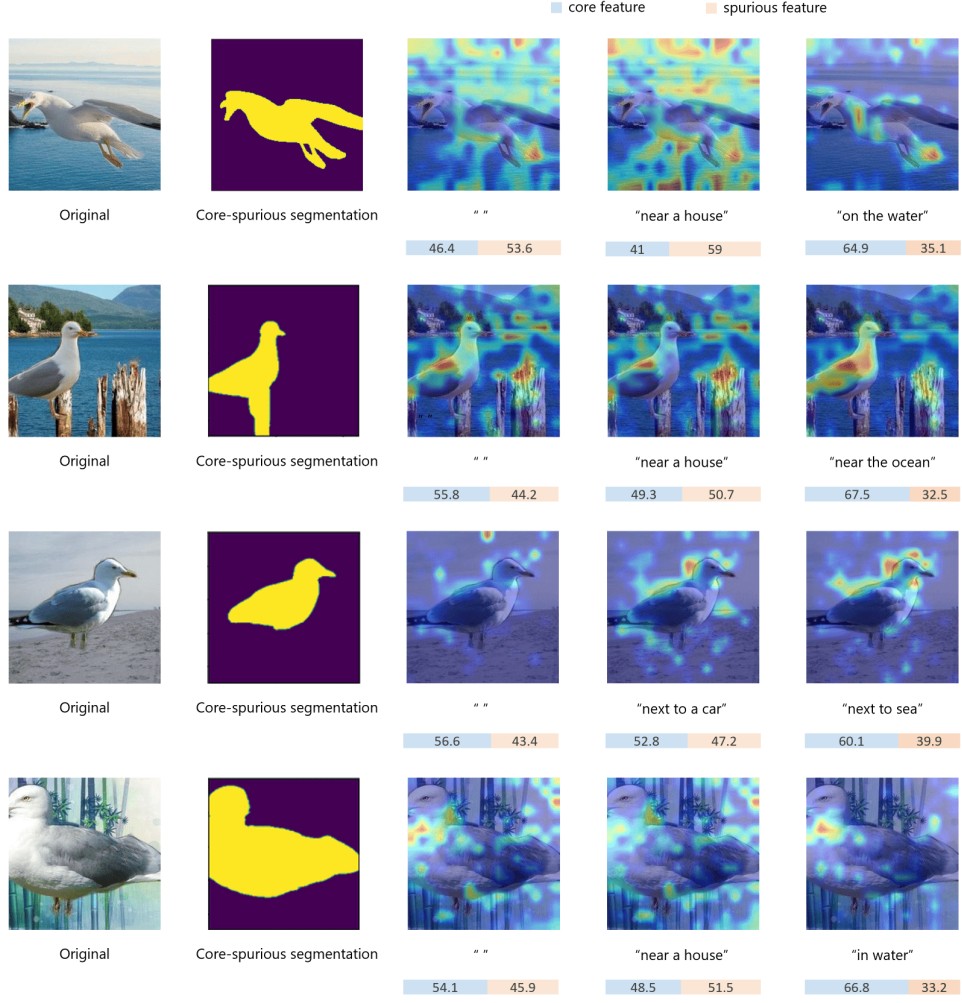

Figure 7: Waterbird images from Waterbirds dataset. Visualization of the original image, the regions of core and spurious features, and the Grad-CAMs obtained using no, incorrect, and ground-truth contextual attributes.

### F.4   DISCOVERING CONTEXTUAL ATTRIBUTES BY LLMS

In this section, we provide an example of how to use GPT-4 (OpenAI, 2023) to generate contextual attributes and their descriptions automatically. We take EuroSAT dataset as an example. There are three steps:

1. Given a dataset (e.g., EuroSAT) with a specific domain, we retrieve similar images (e.g., satellite images) from a large image+text dataset LAION-400M [2].

2. We crawl the captions and randomly sample a limited number of these captions (e.g., 200).

3. We provide GPT-4 with the captions and the information of the dataset, and ask it to extract contextual attributes using Prompt 1.

Table 19 shows the contextual attributes discovered by GPT from captions. Adding those attributes to the domain template, we improve the accuracy from 51.44% to 59.20% (with intervention $\tau = 5$), which is comparable to manually designed ones. However, we found that the attributes identified by GPT are not always appropriate, possibly because of the gap between the retrieved images and our dataset. Future work could involve using image-based searches to find more similar images rather than relying on language-based searches.

Prompt 1: An example prompt for discovering contextual attributes and their descriptions from example captions for EuroSAT dataset. Here we use one description for every attribute value for simplicity.

```
You are a helpful assistant who helps me summarize my text captions. I
    have a dataset of image-caption pairs, where each caption briefly
    describes the image. I need to extract some attributes from these
    textual descriptions that contribute to the data generation process
    of the image.

For example, from the three descriptions ["A black dog on the green grass
    ", "A red car on the road in a bright environment", "A white
    refrigerator"], you can summarize the "background", "color", "
    illumination" as three attributes, with possible values ["grass field
    ", "road", " "], ["black", "green", "red", "white", " "], ["bright",
    "dark", " "] respectively. Note that they each have an empty value
    because the human annotators may choose not to mention them in the
    captions.

Note:
1. The number of potential values for each factor should not exceed 3, so
     you should extract the most representative values.
2. You should summarize at most 5 attributes, covering the most
    representative attributes in the provided captions.
3. I have a list of labels for these images, namely ['annual crop land',
    'forest', 'brushland or shrubland', 'highway or road', 'industrial
    buildings or commercial buildings', 'pasture land', 'permanent crop
    land', 'residential buildings or homes or apartments', 'river', 'lake
     or sea',]. The attributes you summarized should not overlap with the
     concepts of these labels, and the values you summarized should not
    include any of these labels. For example, since "river" is in my
    label set, your summarized values should not include "river" for any
    attributes.
4. The set of all values for all attributes you summarized should not
    overlap.

I need your summary to have the following format:

summerized_factors = {
    "background": [
        "",
        "grass",
```

[2]https://github.com/rom1504/clip-retrieval

```
        "road",
    ],

    "color": [
        "black",
        "green",
        "red",
        "white",
    ],
    "illumination": [
        "bright",
        "dark",
        ""
    ]
}

Here are the captions:

//200 captions
```

Table 19: Contextual attributes and their value descriptions for EuroSAT generated by GPT-4.

| Attributes | Value Descriptions |
|---|---|
| *source* | "", "Yandex satellite", "NASA", "Google Maps" |
| *geographical feature* | "", "island", "ul.", "street" |
| *image type* | "", "satellite", "aerial", "map" |
| *natural phenomenon* | "", "hurricane", "earthquake", "deforestation" |
| *structure type* | "", "residential", "commercial", "fortress" |

## G    IMPACT, LIMITATION AND FUTURE WORK

In this paper, we propose `PerceptionCLIP`, a zero-shot classification method for CLIP that emulates the human visual perception. By doing class inference conditioned on self-inferred contextual attributes, it achieves improved generalization, less reliance on spurious features, and improved interpretability. Along the path of proposing `PerceptionCLIP`, we show CLIP's understanding of object attributes beyond common category features. Our work indicates that CLIP, as a model capable of communicating with humans via natural language, can achieve things that traditional classifiers find challenging. Hence, it still has great potential in zero-shot classification and even broader tasks like image generation. Furthermore, this capability complements the study of neuroscience, enabling a better transition of the latter's research findings into practical use.

**Limitations.** One limitation of `PerceptionCLIP` is its sensitivity to text description perturbations: using different synonyms to describe the same attribute sometimes has non-trivial effects on the results. Although using more descriptions to describe an attribute value (Figure 2) alleviates this sensitivity, this issue is more intrinsic to CLIP and still persists. Future work may overcome this limitation by replacing CLIP with other vision-language models or improving CLIP's sensitivity to textual perturbations (e.g., through training-time text augmentation (Fan et al., 2023)). Another limitation of `PerceptionCLIP` is the need to design a set of contextual attributes. While this process provides a way to integrate human prior knowledge, it also requires additional effort, especially when we aim to cover many attributes. Currently, we use caption retrieval from the LAION-400M dataset and the in-context learning ability of large language models to semi-automate the construction process. In the future, our goal is to automate this process fully. In our paper, we show that directly concatenating multiple attributes' descriptions is a simple and effective way to generate the image's description. Future work can explore more effective and efficient approaches for it.

**Ethical Statement.** In this paper, we use the CelebA dataset, employing descriptors of gender and race to enhance classification accuracy. We acknowledge the sensitivity of these attributes in societal and ethical contexts. Our use of gender and race is strictly as example contextual attributes within our analytical framework, and not as endorsements of any form of racial or gender-based bias. The

inclusion of these attributes is solely for the purpose of exploring and improving the performance of zero-shot classification, without attributing any significance beyond their technical utility. We are committed to maintaining ethical principles in our research and upholding respect and diversity in all aspects of our work.

