# OpenReview forum: "PerceptionCLIP: Visual Classification by Inferring and Conditioning on Contexts"
_ICLR.cc/2024/Conference — ICLR 2024 poster_

### Official Review · Reviewer_VSta · 2023-10-21

**Soundness:** 3 good
**Presentation:** 4 excellent
**Contribution:** 3 good
**Rating:** 8
**Confidence:** 3

**Summary:**

The authors show that adding text that describes the context of an object in an image can improve the performance of CLIP-based zero-shot classification. In addition, they show that said context can be inferred using CLIP itself.

**Strengths:**

1- The paper shows some evidence towards towards the fact that CLIP representations do capture the context as well as the foreground objects, making the alignment with text prompts better when the appropriate context is included in the text.
2- Although the improvements in performance are not large in many of the benchmarks, they come at little cost, probably making it applicable in practice.

**Weaknesses:**

I haven’t found any major weakness in this work (although it is not fully within my expertise).

**Questions:**

Some minor issues:
- For some of the experiments I couldn’t find if they employed ClassAttr or PureAttr.
- In Algo 1, I assume it should be “Set of classes Y” rather than “class Y”, and that the sum is over y \in Y.

---

> ### Author Response · Authors · 2023-11-22
> **Response to Reviewer VSta**
>
> Thanks for your support! We have incorporated all the suggestions from all reviewers into the revised version. Below, we address your questions.
>
> > **Q1:** For some of the experiments I couldn’t find if they employed ClassAttr or PureAttr.
>
> Thank you for highlighting the need for clarity regarding the use of ClassAttr or PureAttr in our experiments. In all the experiments, we use ClassAttr if not specified (noted in Appendix E.4). To make it more clear, we have now included a specific note in Section 7 of our revised paper.
>
> > **Q2:** In Algo 1, I assume it should be “Set of classes Y” rather than “class Y”, and that the sum is over y \in Y.
>
> Thanks for the question. In our paper, we use $Y$ to denote the random variable and $y$ to denote its realization. While, in practice, it represents a set of classes, we opted for the term "class Y" in the algorithm for mathematical conciseness. Your point about the summation is correct. It indeed encompasses all possible realizations.
>
> ---
> Thanks again for your time and effort in reviewing our paper! We are happy to have more discussions for any further questions.

---

### Official Review · Reviewer_vx4m · 2023-10-30

**Soundness:** 2 fair
**Presentation:** 4 excellent
**Contribution:** 3 good
**Rating:** 5
**Confidence:** 4

**Summary:**

The paper explores text-prompting in CLIP, to make the inference process more ``human-like”. The authors show that their two-phased prompting process improves (i) performance, and (ii) robustness against specious features (shortcuts).

**Strengths:**

- In this work, prompting is class-independent; which makes it easily applicable to numerous datasets.
- Authors systematically evaluate the prowess of CLIP in inferring contextual attributes
- The performance gain on domain-specific and out-of-distribution ImageNet datasets shows promise in the claims and approach of the authors
- This work allows for building domain-specific (yet, class independent) augmentations with the possibility of human-in-loop intervention
- The approach presented is elegant and interpretable

**Weaknesses:**

- If I understand correctly, Figure 3 gives only the score (x100) of the correct class in different scenarios. Is this *completely* informative? I think it can be easily misleading. What if the model provides relatively higher scores to some of the wrong classes as well? Can the authors analyse the score distribution of the wrong classes? Reporting the mean of CLIP_score@topK might be a good start to understanding the false positives.
- I appreciate the visualisation study provided by GradCAM as a qualitative analysis but I am not confident of the calculation of “core” versus “spurious” features [1].
- Can the authors also report the random accuracy for Tables 4 and 5? It is important to have a random baseline (that is, random string in place of the inferred context having the same token (and not string) length) here to isolate the effect of “registers” versus actually using the context [1].
- The authors have not provided the code for reproducing the paper; implementation details are missing.

**TL;DR**: The authors make strong claims about reducing the reliance on shortcuts, however, the missing baselines and analyses do not make me confident of their approach. However, some analyses seem misrepresented/miscommunicated. If the authors can answer my questions, I’d be open to changing the score.

Some clarity formatting and typographical errors to rectify:

- Overall, the paper is well-written and ideas well-presented
- The paper, at some points, deviates from standard ICLR formatting:
  - “interleaved” figures and tables (minor inconvenience)
  - Unlabelled table: “Conditioning on ground-truth contextual attributes improves classification accuracy”
- Minor typographical errors:
  - Section 5.1: “Intuitively, we classify an image using both – possible classes and the ground-truth contextual attributes”
  - Remove the asterisk in the author list of “Distributionally robust neural networks“

[1] Darcet, Timothée, Maxime Oquab, Julien Mairal, and Piotr Bojanowski. "Vision Transformers Need Registers." arXiv preprint arXiv:2309.16588 (2023).

**Questions:**

Please refer to weaknesses above

**Details Of Ethics Concerns:**

The authors should consider adding ethical statement (especially since they are using datasets like CelebA with prompts specifically mentioning gender and race).

---

> ### Author Response · Authors · 2023-11-22
> **Response to Reviewer vx4m (1)**
>
> Thanks for your constructive suggestions. They are very helpful in enhancing our paper's quality. We have incorporated all of them to our revised paper. Below, we address your questions in detail.
>
> > **Weakness 1:** If I understand correctly, Figure 3 gives only the score (x100) of the correct class in different scenarios. Is this completely informative? I think it can be easily misleading. What if the model provides relatively higher scores to some of the wrong classes as well? Can the authors analyse the score distribution of the wrong classes? Reporting the mean of CLIP_score@topK might be a good start to understanding the false positives.
>
> We appreciate your insightful comment. To address this, **we've included Figure 5** (referenced alongside Figure 3) to scrutinize the increased CLIP score for both the ground-truth and wrong classes. This analysis is detailed in Appendix E.1.
>
> In Figure 3, we compared {no contextual attribute, correct contextual attribute, wrong contextual attribute, random strings} and found that incorporating correct contextual attribute improves the CLIP score of the ground-truth class the most, indicating CLIP understands those attributes. However, as you rightly pointed out, it does not necessarily equate to increased accuracy since other classes could also have improved CLIP scores. That's why we test the accuracy directly in Section 5.1 and Table 2.
>
> We agree that comparing the improvement for the correct class and the wrong classes is a good idea to complete the picture. Therefore, for each class $y$, we calculate the improvement brought by contextual attributes with
>
> $\Delta_y \triangleq \mathtt{CLIP}(y, z^*; x) - \mathtt{CLIP}_1(y; x)$.
>
> In Figure 5, we contrast these improvements, comparing $\Delta_{y^*}$ for the correct class against $\Delta_{y_{wrong}}$, with the latter being the mean increase across the Top-K incorrect classes. The results confirm that the scores for the correct class see a more substantial increase when the contextual attributes are accurately described. This is because the accurate description of the class and the attribute best aligns with the corresponding image. Figure 3 and 5 together validate that the CLIP model understands the contextual attributes, and describing correct class and attributes yields higher similarity scores as described in Equation (4). Such findings also explain the increased accuracy in Table 2 when incorporating the correct contextual attributes.
>
> > **Weakness 2:** I appreciate the visualisation study provided by GradCAM as a qualitative analysis but I am not confident of the calculation of “core” versus “spurious” features [1].
>
> We appreciate your concern and acknowledge the insights from the concurrent work [1]. The observation of the registers is really interesting and intriguing. Although our visualization, which is based on Grad-CAM, differs from the attention maps tested in their paper, there could also be a register problem in our case. This is an interesting open question. Additionally, whether visualizations like Grad-CAM truly show how the model works remains an open question. Athough these open questions exist, the visualization still provides us with valuable insights, allowing us to better understand the model. We have tried our best to isolate the potential influence of the registers. In Figure 4, we demonstrate that the reduction in reliance on spurious features is notably observed only when the correct context is introduced. This effect is distinctly absent when incorrect contexts or random tokens are used, suggesting a more nuanced interaction than merely an increase in token count. Additionally, we have added a quantitative evaluation in **Table 3 (new results)**. These results align with our visualizations in Figure 4, indicating that the correct contextual information indeed directs the model's focus towards core features, an effect that diminishes with the use of random strings.
>
> While we acknowledge the limitations of current visualization techniques, we believe our multi-faceted approach, combining both qualitative and quantitative analyses, provides a comprehensive understanding of how contextual attributes influence model behavior.
>
> [1] Darcet, Timothée, Maxime Oquab, Julien Mairal, and Piotr Bojanowski. "Vision Transformers Need Registers." arXiv preprint arXiv:2309.16588 (2023).

---

> > ### Author Response · Authors · 2023-11-22
> > **Response to Reviewer vx4m (2)**
> >
> > > **Weakness 3:** Can the authors also report the random accuracy for Tables 4 and 5? It is important to have a random baseline (that is, random string in place of the inferred context having the same token (and not string) length) here to isolate the effect of “registers” versus actually using the context [1].
> >
> > Thank you for this great suggestion. We have added the random baseline to Table 4 and Table 5 (Table 5 and 6 in the revised paper), and discussed the results in Section 7.1 and Appendix E.4.
> >
> > Adding descriptions of contextual attributes to text prompts indeed has two effects: 1) it introduces more tokens to the text prompt, 2) and the tokens describe the contextual attributes. In Figure 3 and Table 2, we have compared the contextual attributes with the same-length random strings to eliminate the first effect.
> >
> > As suggested, we conducted the same **ablation studies in Table 4 and 5 (Table 5 and 6 in the revised paper)**. In Table 4, the ablation studies are on the best attribute compositions for all datasets. In Table 5, we keep the domain template but randomize other contextual attributes. For every dataset, we run 5 times with 5 random seeds and report the mean and variance. Results show that adding random strings can improve the accuracy marginally in most cases, but can also decrease the accuracy as in the cases of EuroSAT and Oxford Pets. More importantly, there is a significant performance gap between using the random strings and the descriptions of contextual attributes, suggesting that the improvement provided by our method primarily stems from the incorporation of contextual attributes.
> >
> >
> > > **Weakness 4:** The authors have not provided the code for reproducing the paper; implementation details are missing.
> >
> > We are happy to provide our code. The code is available at https://anonymous.4open.science/r/PerceptionCLIP-B351.
> >
> > > **Format and typos. Need an ethical statement.**
> >
> > Thanks for pointing them out. We have adjusted the format, corrected the typos, and added an ethical statement in Appendix G.
> >
> > ---
> >
> > Thanks again for your time and effort in reviewing our paper! Could you please consider increasing the score if you are satisfied with our answers? We are happy to have more discussions for any further questions.

---

> > > ### Comment · Reviewer_vx4m · 2023-11-23
> > >
> > > Thank you authors for answering my queries. While I am satisfied with some clarifications, I believe that some experiments uncovered more serious doubts.
> > >
> > > **W1**: While the increase in the correct class is substantial, the mean increase in the wrong classes is also not trivial. Ideally, wouldn’t you expect this value to be negative? I am not sure how $\Delta_{y_{wrong}}$ supports the author's results and claims on robustness. Is it that the CLIP in-turn develops a stronger context-bias (say because the context attribute is “on water”, the model has higher prediction scores for other classes that co-occur with water as well). **If such might be the case, which the authors haven’t ruled out and seems apparent from the newer results, the work might be creating a new problem while solving one.**
> > >
> > > **W2**: My doubts regarding the speciousness of the methodology still stand true. While the qualitative results are definitely interesting, they are not convincing enough to extrapolate the robustness claims
> > >
> > > **W3**: The results are quite nice! Appreciate their incorporation in the main paper
> > >
> > > Thanks for providing the code and well-organised discussion. The authors’ response to W1 has left me more skeptical of the claims of the paper.
> > >
> > > I am open to hearing the authors' response but will keep the ratings as it is till then.

---

### Official Review · Reviewer_U13E · 2023-11-01

**Soundness:** 2 fair
**Presentation:** 3 good
**Contribution:** 3 good
**Rating:** 5
**Confidence:** 4

**Summary:**

Inspired by the human perception process that the contextual attributes are separated from the foreground objects, this paper proposes a training-free, two-step zero-shot classification method PerceptionCLIP to first infer the contextual attributes (e.g., background) and then performs object clas- sification conditioning on them. A proof-of-concept investigations reveal that conditioning on ground-truth contextual attributes improves CLIP’s zero-shot classification. The proposed PerceptionCLIP demonstrates performance gain and improved interpretability on several datasets.

**Strengths:**

The idea of imitating human perception process to improve the generalization and group robustness of the image classification model is insightful for the community.
The proposed method is extensively evaluated on 11 datasets.

**Weaknesses:**

1. Collecting the contextual attributes requires either pre-knowledge for the test image or a large dataset containing captions, which hinders the generalization ability of the proposed method in the real world. For instance, contextual attributes for the CelebA dataset are manually defined, e.g., gender, age, etc. To collect the contextual attributes for the remote sensing dataset EuroSAT, the authors first retrieve similar images and captions from a large image+text dataset LAION-400M, then ask GPT-4 to summarize the contextual attributes. What if we do not have external datasets to provide captions?

2. The qualitative results in Figure 4 indicate that introducing the contextual attributes reduces reliance on the spurious features and the model focuses more on the core features. It would be fairer to provide a quantitative evaluation, e.g., counting the percentage of model attention on the core features versus on spurious feature on all test set in ImageNet, and compare the ratio of different models.

3. The performance gain seems marginal on most of the datasets. For instance, in Table 4, the performance gain is only around 2% on ImageNet, ImageNetV2, ImageNet-A and ImageNet-Sketch. Besides, since introducing a random attribute or even a wrong attribute can improve the accuracy in Table 2, it would be interesting to include the results of the wrong attribute and random attribute in Table 4 as well.

4. The results in Table 7 are not consistent among different backbones. It is hard to get any conclusion on which method is better.

**Questions:**

In Table 7, why lower gap between the Avg and Worst is better?

---

> ### Comment · Reviewer_U13E · 2023-11-22
>
> Since the authors did not provide responses, I would like to keep the original rating.

---

> > ### Author Response · Authors · 2023-11-22
> > **Sorry about the delay**
> >
> > Sorry about the delay. We are diligently running new experiments to strengthen the paper as suggested by reviewers. We will post the updated manuscript, experimental results, as well as detailed response shortly.
> >
> > Thank you very much for your service and for engaging in the discussion!
> >
> > best regards,
> > authors

---

> > ### Author Response · Authors · 2023-11-22
> > **Apology for Delay in Rebuttal Response**
> >
> > Dear Reviewer U13E,
> >
> > We sincerely apologize for the delay in our rebuttal. We have posted our response to your questions and are in the process of sequentially posting our response to other reviewers including a general response.
> >
> > Thanks for your patience!
> >
> > Best regards,
> > Authors

---

> ### Author Response · Authors · 2023-11-22
> **Response to Reviewer U13E (1)**
>
> Thank you for the constructive feedback. Your suggestions are very helpful. We have incorporated all of them in our revised paper. Here, we address your questions and concerns in detail.
>
> > **Weakness 1:** Collecting the contextual attributes requires either pre-knowledge for the test image or a large dataset containing captions, which hinders the generalization ability of the proposed method in the real world. For instance, contextual attributes for the CelebA dataset are manually defined, e.g., gender, age, etc. To collect the contextual attributes for the remote sensing dataset EuroSAT, the authors first retrieve similar images and captions from a large image+text dataset LAION-400M, then ask GPT-4 to summarize the contextual attributes. What if we do not have external datasets to provide captions?
>
> Thanks for bringing this up. It's true that our method, like other zero-shot classification approaches, requires some prior knowledge of the data. However, the type and amount of prior knowledge needed in our case are relatively minimal and practical to obtain. We only need **basic, class-independent** contextual attributes like background or illumination for natural images and gender for facial images, which are generally known or easily inferred. This simplicity contrasts with other methods [1-2], which require more **detailed, class-specific** descriptions for every class.
>
> One advantage of our method lies in its flexibility with contextual attributes. We only need a set of possible contextual attributes since CLIP itself can first infer the most relevant contextual attribute from this set. Such flexibility makes our approach adaptable to a wide range of datasets which was demonstrated in our paper across 13 different datasets.
>
> For situations where contextual attributes are not readily available, we propose an **alternative**: using the LAION caption dataset and GPT-4 to automatically identify relevant attributes. This approach is grounded in the fact that OpenCLIP is trained on LAION-2B [3], and CLIP was trained on similar large-scale caption datasets. Since CLIP acquires the knowledge of contextual attributes from the training data, LAION is a good source to find contextual attributes automatically for almost all datasets.
>
> [1] Menon, Sachit, and Carl Vondrick. "Visual classification via description from large language models." ICLR 2023.
>
> [2] Pratt, Sarah, et al. "What does a platypus look like? generating customized prompts for zero-shot image classification." CVPR 2023.
>
> [3] https://laion.ai/blog/large-openclip/
>
> > **Weakness 2:** The qualitative results in Figure 4 indicate that introducing the contextual attributes reduces reliance on the spurious features and the model focuses more on the core features. It would be fairer to provide a quantitative evaluation, e.g., counting the percentage of model attention on the core features versus on spurious feature on all test set in ImageNet, and compare the ratio of different models.
>
> Thank you for your valuable suggestion. **We've now added Table 3 in our paper**, presenting quantitative results. This table, along with a detailed discussion in Section 5.1 and Appendix E.3, addresses this point.
>
> To calculate the percentage of model attention on the core versus spurious feature, we need the segmentation of them. We do not have such segmentation for the ImageNet dataset. Therefore, our evaluation is conducted on the Waterbirds dataset where we have the segmentation of the core feature (e.g., bird) and spurious feature (e.g., background).
>
> The results in Table 3 highlight that when the correct context (here, background) is specified, CLIP shifts its focus more toward the core feature for classification. This effect is not observed when using incorrect or random context. Thus, both Figure 4 and Table 3 collectively demonstrate that introducing contextual attributes effectively reduces the model's reliance on spurious features and enhances its focus on core features.

---

> > ### Author Response · Authors · 2023-11-22
> > **Response to Reviewer U13E (2)**
> >
> > > **Weakness 3.1.:** The performance gain seems marginal on most of the datasets. For instance, in Table 4, the performance gain is only around 2% on ImageNet, ImageNetV2, ImageNet-A and ImageNet-Sketch.
> >
> > We appreciate your observation of the performance gains. It's important to highlight that our method operates within the **zero-shot classification paradigm** without any additional training involved, therefore **even modest improvements are significant milestones**. In our study, spanning Tables 4 and 5 (Table 5 and 6 in the revised paper), we evaluated our approach across 11 diverse datasets. Notably, we observed an average improvement of 3% - **a substantial enhancement in the zero-shot context**. Significant gains were evident on datasets like EuroSAT (7.79% increase), Flowers102 (5.13%), and ImageNet-R (4.39%). These results underline the efficacy of incorporating contextual attributes into the CLIP model.
> >
> > Our goal extends beyond just achieving state-of-the-art results. We aim to provide the community with **an interesting and intriguing finding** - CLIP knows the contextual attributes and incorporating them helps zero-shot classification. The comprehensive analyses in Sections 4 and 5, coupled with these promising outcomes, solidify our findings. We believe that this research offers valuable insights into the capabilities of vision-language models, shedding light on their potential in classification and even broader areas like image generation. It's our hope that these findings will **inspire further exploration and development** in the field.
> >
> > > **Weakness 3.2:** Besides, since introducing a random attribute or even a wrong attribute can improve the accuracy in Table 2, it would be interesting to include the results of the wrong attribute and random attribute in Table 4 as well.
> >
> > Thanks for this great suggestion. We have **added the random baseline to Table 4 and Table 5 (Table 5 and 6 in the revised paper)**, and discussed the results in Section 7.1 and Appendix E.4.
> >
> > Adding descriptions of contextual attributes to text prompts indeed has two effects: 1) it introduces more tokens to the text prompt, 2) and the tokens describe the contextual attributes. In Figure 3 and Table 2, we have compared the contextual attributes with the wrong attributes and same-length random strings to eliminate the first effect.
> >
> > As suggested, we conducted the same ablation studies in Table 4 and 5 (Table 5 and 6 in the revised paper). Unlike the cases in Figure 3 and Table 2 where we manually change the attributes, we do not know what are correct (or wrong) attributes for cases in Table 4 and 5. Therefore, we only tested random attributes. In Table 4, the ablation studies are on the best attribute compositions for all datasets. In Table 5, we keep the domain template but randomize other contextual attributes. For every dataset, we run 5 times with 5 random seeds and report the mean and variance. Results show that adding random strings can improve the accuracy marginally in most cases (the potential reasons are discussed in Appendix E.4), but can also decrease the accuracy as in the cases of EuroSAT and Oxford Pets. More importantly, there is a significant performance gap between using the random strings and the descriptions of contextual attributes, suggesting that the improvement provided by our method primarily stems from the incorporation of contextual attributes.

---

> ### Author Response · Authors · 2023-11-22
> **Response to Reviewer U13E (3)**
>
> > **Weakness 4 & Question 1:** The results in Table 7 are not consistent among different backbones. It is hard to get any conclusion on which method is better. In Table 7, why lower gap between the Avg and Worst is better?
>
> In Table 7 (Table 8 in our revised paper), our goal is to demonstrate how our method enhances group robustness by incorporating contextual attributes regardless of the backbones. Group robustness measures how consistently the model performs across different groups within a dataset. For example, in the Waterbirds dataset, there are four distinct groups: {waterbird on water, landbird on land, waterbird on land, landbird on water}. The first two are considered majority groups as they are more prevalent in the training data, while the latter two are minority groups. Typically, due to training set imbalances, models tend to perform better in majority groups. However, we need models that perform well in all groups. To this end, following the convention in this field [1], we assess model accuracy on each of the four groups, calculating both the average accuracy (Avg) and the worst group accuracy (Worst). The gap (Gap) is the difference between these two metrics. A higher gap indicates a bias towards certain groups, implying less consistency and robustness. Conversely, a lower gap signifies a more balanced and robust model performance across all groups. Therefore, the lower gap is better.
>
> We would like to clarify that the purpose of Table 7 isn't to compare different backbones. Instead, we aim to show that our method is **effective across various backbones**. By comparing our method (with $\mathcal{Z}$) against the baseline (without $\mathcal{Z}$) for each backbone, we demonstrate that our approach consistently reduces the gap between the average and worst group accuracies, thereby achieving better group robustness.
>
> [1] Sagawa, Shiori, et al. "Distributionally robust neural networks for group shifts: On the importance of regularization for worst-case generalization." ICLR 2020.
>
> ---
> Thanks again for all your time and effort in reviewing our paper! Could you please consider increasing the score if you are satisfied with our answers? We are happy to have more discussions for any further questions.

---

### Official Review · Reviewer_k2xX · 2023-11-03

**Soundness:** 3 good
**Presentation:** 3 good
**Contribution:** 3 good
**Rating:** 6
**Confidence:** 4

**Summary:**

This paper is inspired by human visual perception, where humans first discern contextual attributes, such as background and orientation, to distinguish objects, and then classify them. Similarly, when CLIP is provided with these contextual attributes, it improves in zero-shot image classification and reduces dependence on irrelevant features. Authors found that CLIP can deduce these attributes from an image itself and based on this fact to propose PerceptionCLIP. PerceptionCLIP first determines contextual attributes from an image and then classifies the object based on these attributes. Experiments are done on CLIP's zero-shot classification settings and show clear improvements over the original CLIP.

**Strengths:**

- The paper is well written and easy-to-follow. While the concept of utilizing background information for image classification isn't groundbreaking in literature, its application to CLIP could be innovative.
- The experiments show clear advantages of using contextual attributes over the traditional 80 templates.

**Weaknesses:**

- The authors assert at least twice that PerceptionCLIP mirrors human perception. However, I'm not entirely convinced. Authors gave the preliminary that: “humans first infer contextual attributes (e.g., background and orientation) which help separate the foreground object from the background, and then classify the object based on this information.” Yet, there's no evidence indicating that PerceptionCLIP actively separate foreground from background during classification, or that such separation is utilized the model. It's possible that PerceptionCLIP utilizes background attributes differently.
- The authors refer to background information as spurious features (e.g. Figure 1). To my knowledge, it is not completely correct. Though they can sometimes overlap, they are not the same. Background information is a broader concept, while spurious features specifically refer to misleading patterns that a model might incorrectly learn as being important. In addition, the GradCAM in Figure 1 primarily emphasizes the foreground, consistent with [a], without highlighting any reduced reliance on spurious features. It's more accurate to state that it offers enhanced focus on foreground objects.
- When I like the idea of Textual descriptions for contextual attributes Sec 4.1., I could not find how exactly you map Z using the proposed annotation function alpha to attribute text descriptions. Also, why do you say this annotation function model human preferences in captioning? Authors may also want to clarify the p value associated with the textual descriptions.  I imagine that these descriptions can also easily obtained using LLMs (e.g. ChatGPT).
- The name of the proposed metric can be easily confused with the original CLIP score. How about naming it as Attribute-CLIP.

**Questions:**

Post-rebuttal:

I genuinely appreciate your great efforts put into this rebuttal!

I read the (updated) paper one more time, the authors' responses to me thoroughly, and the responses to other reviewers.
While PercentionCLIP is indeed powerful and but answer of "why does it work" is not fully addressed via the GradCAM visualization as raised by W2 of reviewer vx4m.

I would like to keep my rating for now and may change later after discussing with other reviewers.

---

> ### Author Response · Authors · 2023-11-22
> **Response to Reviewer k2xX (1)**
>
> Thank you for your support and all your questions. Here, we address them in detail.
>
> > **Weakness 1:** The authors assert at least twice that PerceptionCLIP mirrors human perception. However, I'm not entirely convinced. Authors gave the preliminary that: “humans first infer contextual attributes (e.g., background and orientation) which help separate the foreground object from the background, and then classify the object based on this information.” Yet, there's no evidence indicating that PerceptionCLIP actively separate foreground from background during classification, or that such separation is utilized the model. It's possible that PerceptionCLIP utilizes background attributes differently.
>
> Thanks for your feedback! Let me clarify how PerceptionCLIP emulates human perception. Our approach, outlined in Algorithm 1, involves two main steps inspired by the human perceptual process. Firstly, CLIP infers contextual attributes, which is akin to how we humans first discern context, like background or orientation, to understand a scene. Secondly, this contextual information is used for classification, much like how we classify objects based on the context we've perceived.
>
> It's important to note that in our model, separating foreground from background is just one example of using contextual attributes, not the entire story. The key aspect of human perception that we're trying to capture is the ability to use context effectively. By integrating these contextual attributes into our model's prediction process, we've seen a notable boost in zero-shot classification performance. PerceptionCLIP isn't about mimicking the exact human process of separating foreground and background, but rather about leveraging context in a way that's inspired by human perception.
>
> > **Weakness 2:** The authors refer to background information as spurious features (e.g. Figure 1). To my knowledge, it is not completely correct. Though they can sometimes overlap, they are not the same. Background information is a broader concept, while spurious features specifically refer to misleading patterns that a model might incorrectly learn as being important. In addition, the GradCAM in Figure 1 primarily emphasizes the foreground, consistent with [a], without highlighting any reduced reliance on spurious features. It's more accurate to state that it offers enhanced focus on foreground objects.
>
> Thank you for pointing out the distinction between spurious features and background information. You're right in noting that spurious features are generally misleading patterns that a model might erroneously learn as significant. In our context, we use the term "spurious feature" to also encompass parts of the image with a recognizable concept, like the background, as referenced in several works including [1].
>
> In Figure 1, our aim was to demonstrate the model's shift from relying on spurious (or background) features to focusing more on core (or foreground) objects. The model's reliance ratio changes from 31:69 (background vs foreground) in CLIP to 23:77 in PerceptionCLIP, suggesting that our method indeed reduces dependency on what we've termed as spurious features. This aligns with your observation about the model offering an enhanced focus on foreground objects. We appreciate your feedback and will consider clarifying this distinction in our revision.
>
> [1] Moayeri, Mazda, et al. "Spuriosity Rankings: Sorting Data to Measure and Mitigate Biases." NeurIPS 2023.

---

> > ### Author Response · Authors · 2023-11-22
> > **Response to Reviewer k2xX (2)**
> >
> > > **Weakness 3.** When I like the idea of Textual descriptions for contextual attributes Sec 4.1., I could not find how exactly you map Z using the proposed annotation function alpha to attribute text descriptions. Also, why do you say this annotation function model human preferences in captioning? Authors may also want to clarify the p value associated with the textual descriptions. I imagine that these descriptions can also easily obtained using LLMs (e.g. ChatGPT).
> >
> > Thank you for your interest in our approach to mapping contextual attributes to textual descriptions. As illustrated in Figure 2, our annotation function $\alpha$ serves as a mapping from each attribute value, like upright, to a corresponding pre-defined distribution of descriptive terms. These terms can include direct descriptions such as "upright" or "upstanding" and no description (""). The choice of these distributions aims to mirror human preferences in captioning. This assumption is based on the nature of the CLIP model, which is trained on (image, caption) pairs from a web-scale dataset like LAION [2]. Although the exact training data of CLIP is unknown, we infer that it broadly reflects human captioning preferences due to its web-derived nature.
> >
> > In practice, since determining precise human preferences is challenging, we opted for a uniform distribution approach for simplicity, using two to four descriptions per attribute. This simple method has already yielded significant results, suggesting its effectiveness. Your suggestion of utilizing p-values from LLMs like ChatGPT to assign probabilities to different descriptions is intriguing. This could potentially refine the alignment with human preferences and enhance our method's performance. We acknowledge the potential of this approach and plan to explore it in future research. For the current scope of our work, we have maintained a simpler model, which has already demonstrated promising outcomes.
> >
> > [2] https://laion.ai/blog/large-openclip/
> >
> > > **Weakness 4.** The name of the proposed metric can be easily confused with the original CLIP score. How about naming it as Attribute-CLIP.
> >
> > Thanks for your suggestion regarding the metric's name. We understand your concern about potential confusion with the original CLIP score. Our version, denoted as $\mathtt{CLIP}(y, z; x)$, maintains the fundamental calculation of the original CLIP score, with the key distinction being the incorporation of contextual attributes into the text prompt, represented by $z$ in the notation.
> >
> > We believe that keeping the name close to the original CLIP score underscores this continuity and the foundational similarities between the two. While we're open to considering a name change like 'Attribute-CLIP' in future iterations, we feel that retaining the current name for now will help minimize confusion, especially for other reviewers. We appreciate your input and will take it into account as we continue to develop our work.
> >
> > ---
> >
> > Thanks again for your time and effort in reviewing our paper! We are happy to have more discussions for any further questions.

---

### Author Response · Authors · 2023-11-22
**General Response**

$\newcommand{kxx}{\textcolor{red}{\mathrm{k2xX}}}$
$\newcommand{vxm}{\textcolor{blue}{\mathrm{vx4m}}}$
$\newcommand{vsta}{\textcolor{green}{\mathrm{VSta}}}$
$\newcommand{ue}{\textcolor{orange}{\mathrm{U13E}}}$

We thank all the reviewers for their constructive feedback and insightful questions. We are encouraged that the reviewers find our paper well-written ($\kxx$, $\vxm$, $\vsta$), and our method sound and insightful for the community ($\kxx$, $\ue$, $\vxm$, $\vsta$). All the suggestions are very helpful in enhancing the quality of our paper. We have addressed all the suggestions and questions in our revised paper. The newly added content is highlighted in blue.

## Paper Updates and New Experiments

Here, we summarize two major revisions.

1. **In Table 5 and 6, we have added the random baselines.** We also discussed the results in Section 7.1 and Appendix E.4. The comparison between ours and random baselines indicates that the effectiveness of our method primarily stems from contextual attributes. (Thanks for Reviewer $\ue$ and $\vxm$'s suggestion)
2. **We added Table 3 for a quantitative evaluation.** We also discussed the results in Section 5.1 and Appendix E.3. The results in Table 3 highlight that when the correct context is specified, CLIP shifts its focus more toward the core feature for classification. This effect is not observed when using incorrect or random context. Thus, both Figure 4 and Table 3 collectively demonstrate that introducing contextual attributes effectively reduces the model's reliance on spurious features and enhances its focus on core features. (Thanks for Reviewer $\ue$'s suggestion and Reviewer $\vxm$'s question)

Other revisions include:

3. **We added Figure 5 in Appendix E.1** as an ablation of Figure 3. (Thanks for Reviewer $\vxm$'s suggestion)

4. **We added an ethical statement in Appendix G.**

5. We refined the format and corrected the typos.

6. We shared our code here https://anonymous.4open.science/r/PerceptionCLIP-B351

## Summary of Novelty and Contributions
We would like to reiterate the following key points:
1. Our paper is the **first one** that finds CLIP understands the contextual attributes and leveraging them notably improves the zero-shot classification.
2. We propose an easy to implement two-step method to infer and leverage contextual attributes. Our method **requires no training**.
3. Our findings and method are validated by comprehensive analyses in Section 4-5 and experiments on **13 datasets** in Section 7.
4. We believe our findings and observations are **insightful for the community**.
---
Again, we would like to extend our sincere gratitude to all reviewers for their valuable time and effort!

---

### Meta-Review · Area_Chair_7csU · 2023-12-07

**Metareview:**

This paper proposes a zero-shot visual classification approach called PerceptionCLIP which infers and conditions on contextual attributes.  Four reviewers provided ratings of 6, 5, 6, 8 (these scores reflect one of the reviewers who commented that they are increasing their score from 5 to 6 but did not change it in their ratings).  Positive points about the paper include the sound approach, extensive experiments, promising results, and clear writing.  Negatives points include concerns regarding some claims (e.g., connections to human perception), question on why the method works, marginal improvements over baselines, and some missing baselines and analyses.  The authors provided a detailed rebuttal including new experiments to try to address many of these concerns.  The reviewers also engaged in extensive discussion (excluding the one reviewer who gave a 5, who did not respond to the authors' rebuttal). Many of the concerns were addressed by the rebuttal, but a remaining concern was the question of "why the approach works" not being fully addressed.  The paper, rebuttal, discussion, and author messages were carefully discussed among the ACs, and the ACs feel that the positives outweigh the negatives, and hence recommend acceptance.

**Justification For Why Not Higher Score:**

There is a remaining concern, and there is not strong support for a higher rating by the reviewers.  The reviewer who gave the highest score of 8 is not very confident as indicated in their review as well as in their message to the ACs.

**Justification For Why Not Lower Score:**

The positives outweigh the negatives, and the paper can make a good contribution to the community.

---

### Decision · Program_Chairs · 2024-01-16

Accept (poster)